# Idealized 3D Auxetic Mechanical Metamaterial: An Analytical, Numerical, and Experimental Study

**DOI:** 10.3390/ma14040993

**Published:** 2021-02-20

**Authors:** Naeim Ghavidelnia, Mahdi Bodaghi, Reza Hedayati

**Affiliations:** 1Department of Mechanical Engineering, Amirkabir University of Technology (Tehran Polytechnic), Hafez Ave, Tehran 1591634311, Iran; n.ghavidelnia@aut.ac.ir; 2Department of Engineering, School of Science and Technology, Nottingham Trent University, Nottingham NG11 8NS, UK; mahdi.bodaghi@ntu.ac.uk; 3Novel Aerospace Materials, Faculty of Aerospace Engineering, Delft University of Technology (TU Delft), Kluyverweg 1, 2629 HS Delft, The Netherlands

**Keywords:** 3D printing, mechanical metamaterial, auxetics, negative Poisson’s ratio, 3D re-entrant

## Abstract

Mechanical metamaterials are man-made rationally-designed structures that present unprecedented mechanical properties not found in nature. One of the most well-known mechanical metamaterials is auxetics, which demonstrates negative Poisson’s ratio (NPR) behavior that is very beneficial in several industrial applications. In this study, a specific type of auxetic metamaterial structure namely idealized 3D re-entrant structure is studied analytically, numerically, and experimentally. The noted structure is constructed of three types of struts—one loaded purely axially and two loaded simultaneously flexurally and axially, which are inclined and are spatially defined by angles θ and φ. Analytical relationships for elastic modulus, yield stress, and Poisson’s ratio of the 3D re-entrant unit cell are derived based on two well-known beam theories namely Euler–Bernoulli and Timoshenko. Moreover, two finite element approaches one based on beam elements and one based on volumetric elements are implemented. Furthermore, several specimens are additively manufactured (3D printed) and tested under compression. The analytical results had good agreement with the experimental results on the one hand and the volumetric finite element model results on the other hand. Moreover, the effect of various geometrical parameters on the mechanical properties of the structure was studied, and the results demonstrated that angle θ (related to tension-dominated struts) has the highest influence on the sign of Poisson’s ratio and its extent, while angle φ (related to compression-dominated struts) has the lowest influence on the Poisson’s ratio. Nevertheless, the compression-dominated struts (defined by angle φ) provide strength and stiffness for the structure. The results also demonstrated that the structure could have zero Poisson’s ratio for a specific range of θ and φ angles. Finally, a lightened 3D re-entrant structure is introduced, and its results are compared to those of the idealized 3D re-entrant structure.

## 1. Introduction

The Poisson’s ratio is defined as the ratio of the negative value of transverse strain to the longitudinal strain of a body subjected to uniaxial stress [1]. The Poisson’s ratio is usually assumed to be positive (v>0) for natural materials, which means that the material contracts transversely when stretched longitudinally or expands transversely when compressed longitudinally [2]. In contrast, there are some materials that reveal the opposite behavior of common materials and present negative Poisson’s ratio, which means that their transverse dimensions increase (or decrease) when they are subjected to axial tensile (or compressive) loads [3]. These materials have been given different names in the literature, including anti-rubber, dilational [4], and auxetic materials [5]. The term “auxetic” is currently the most widely accepted name used for materials with a negative Poisson’s ratio. Auxetic behavior has been observed in a few natural materials such as cristobalite (polymorphic silicones) [6], metals [7], zeolites [8], silicates [9,10], and some biological tissues such as cancellous bone [11], tendons [12], and some animal skins [13,14].

Mechanical metamaterials are man-made materials with unusual mechanical properties and special functionalities not found in nature [15,16,17,18,19]. Negative Poisson’s ratio could be considered as one of such special mechanical properties [20,21,22,23,24,25,26,27,28,29,30,31]. Due to recent advances in additive manufacturing techniques (3D printing), which provides the possibility of fabrication of rationally-designed materials with the desired accuracy in nano/micro-scale [32,33,34,35], the studies concentrated on design and analyzing auxetic metamaterials have been increasing exponentially during the last few years [36,37,38,39,40]. Different geometries with different deformation mechanisms have been introduced for auxetic structures. One of the first auxetic structures was proposed by Roberts [21], the first auxetic foam was presented by Lakes [41], and the first 2D molecular auxetic models (spontaneously forming auxetic phases) were studied by Wojciechowski [22,23]. The term auxetic in the scientific literature was introduced by Evans [24]. The re-entrant hexagonal honeycomb structure is one of the most well-known open-cell auxetic structures [42]. In several studies [43,44,45,46], the effect of relative density and re-entrant angle variations on mechanical properties such as Poisson’s ratio and elastic modulus of typical 2D re-entrant structure in the elastic and plastic range have been analyzed for predicting the behavior of the structure. To improve the mechanical properties such as strength and auxeticity of the re-entrant structures and for answering the new requirements in the industry, some researchers have introduced and built novel 3D re-entrant structures inspired by typical 2D re-entrant unit cells and have established analytical and numerical solutions for them [47,48,49,50].

The unit cell presented by Yang et al. in 2015 [50] (Figure 1a) is one of the successful 3D auxetic structures, which has been solved analytically in two directions and evaluated by experimental and numerical results. In another study, Wang et al. (2016) [51] proposed an interlocking assembly manufacturing technique for fabricating the re-entrant lattice structure proposed by Yang et al. (2015) [50] but with a shifted unit cell (Figure 1b), which showed great advantages over additive manufacturing technique. Chen et al. (2017) and Xue et al. (2020) [52,53] applied some modifications to the noted 3D re-entrant auxetic structure by adding some struts to enhance the mechanical properties of the lattice structure, and in particular, to increase the stiffness of this auxetic structure (Figure 1c,d). Although the 3D reentrant structures studied in the noted works are successful auxetic structures for many applications, they still lack some geometrical characteristics, which could raise problems in many applications. First of all, even though the noted 3D re-entrant unit cells are symmetrical in the lateral directions, but their geometry is different in the vertical (i.e., loading) direction. Therefore, they exhibit dissimilar mechanical properties in the three main orthogonal directions, which means that they are not truly 3D auxetic structures. Second, the struts at the edge of the noted unit cells are shared by adjacent unit cells in the lattice structure. Since the auxetic structures are useful in many applications (such as biomedical implants [54,55], sandwich panels [56], etc.) that require graded micro-structures, the noted shared struts cause huge complexities and difficulties in constructing truly graded lattice structures. Hence, there remains a gap to propose an ideal unit cell that could overcome these deficiencies of typical 3D re-entrant structure.

In this study, a specific type of mechanical metamaterial, namely, an idealized 3D re-entrant structure [41,57] constructed of three strut types, one type loaded purely axially and two types loaded both flexurally and axially, which are inclined and are spatially defined by angles θ and φ (see Figure 2), is studied to obtain analytical relationships for its mechanical properties. Analytical solutions based on two well-known beam theories namely Euler–Bernoulli and Timoshenko theories are derived. Moreover, two numerical methods one based on volumetric elements and one based on beam elements are implemented for validating the results. A definition for strut effective length has been presented in order to simulate what the struts experience in real-life conditions, and analytical results have been presented based on the effective length. Moreover, a lightened auxetic unit cell extracted from the general re-entrant unit cell (presented in Figure 2) is introduced and its mechanical properties are compared to those of the ideal re-entrant structure. Several specimens have also been additively manufactured and tested under compressive loading conditions. Finally, the results of the derived analytical and constructed numerical models and the manufactured specimens are compared to one another.

## 2. Materials and Methods

### 2.1. Analytical Solution

#### 2.1.1. Relative Density

Each re-entrant unit cell (Figure 2a and Appendix A) consists of three types of struts with lengths $l_{1},\;l_{2},\;l_{3}$ and cross-sectional areas $A_{1},\;A_{2},\;A_{3}$. The unit cell is fully defined with two unique angles θ and φ, the Type-I strut length, and the main cube size (the main cube is the hypothetical cube formed by connecting vertices $D_{i}$ where i=1 to 8). Angle θ is defined as the angle between Type-II strut and the face diagonals of the main cube, and angle φ is defined as the angle between a Type-III strut and the corresponding side of the main cube (Figure 2b). Each re-entrant unit cell with mentioned strut lengths ($l_{1},\;l_{2},\;l_{3})$ and characteristic angles (θ, φ) occupies a cubic volume with side length of $2\left(l_{1}+l_{3}\cos\varphi -l_{2}\sin\theta \right)$ and hence the volume of $V_{total}=8{\left(l_{1}+l_{3}\cos\varphi -l_{2}\sin\theta \right)}^{3}$. Because the unit cell is made up of six struts of Type I, 24 struts of Type II, and 24 struts of Type III, the volume occupied by all struts inside the unit cell is $V_{strut}=6\pi r_{1}^{2}l_{1}+24\pi r_{2}^{2}l_{2}+24\pi r_{3}^{2}l_{3}$. Therefore, according to the basic definition of relative density $\mu =\frac{V_{strut}}{V_{total}}$, the relative density of a re-entrant unit cell is given by
(1)$$\mu =\frac{6A_{1}l_{1}+24A_{2}l_{2}+24A_{3}l_{3}}{L_{unitcell}{}^{3}}=\frac{3\pi \left(r_{1}^{2}l_{1}+4r_{2}^{2}l_{2}+4r_{3}^{2}l_{3}\right)}{4{\left(l_{1}+l_{3}\cos\varphi -l_{2}\sin\theta \right)}^{3}}$$

#### 2.1.2. Stiffness Matrix

In this subsection, analytical relationships for mechanical properties of the 3D re-entrant unit cell such as elastic modulus, Poisson’s ratio, and yield stress are derived as functions of elastic properties of the bulk material ($E_{s},G_{s},\;\sigma_{ys},\;$and $v_{s}$) and the geometrical dimensions of the unit cell. The re-entrant unit cell has symmetry with respect to its major planes (see Appendix A). Therefore, studying 1/8 of the unit cell is sufficient for obtaining the mechanical properties of the unit cell and, hence, the corresponding lattice structure. Due to the intrinsic symmetries in the geometry of the re-entrant unit cell and the applied boundary conditions on the unit cell, seven unique group of vertices can be recognized, which are denoted by letters A, B, C, D, E, F, and G, as shown in Figure 2a. For the sake of brevity and as the vertices named by similar letters demonstrate identical behaviors, in many cases, instead of referring to a vertex as $X_{i}$, we refer to it as X (*X* representing A, B, C, D, E, F, and G). For instance, in many cases, we use the name E instead of $E_{1}$.

If the origin of the coordinate system is considered to be located in the midst of the unit cell, each re-entrant unit cell has four vertical (XY, YZ, and two bisectors of XY and YZ) and one horizontal (XZ) symmetry planes (as shown in Appendix A). All the vertices located in the symmetry planes are only allowed to translate and rotate in their corresponding planes. Vertices $A_{i}$ and $B_{i}$ (for *i* = 1, 2) lie in all of the four vertical symmetry planes, so they are only allowed to translate in the Y direction, and they are constrained rotationally. Therefore, they create two degrees of freedom (DOFs) $q_{8}$ and $q_{7}$ at vertices $A_{i}$ and $B_{i}$, respectively. Vertices $C_{i}$ (for i=1 to 8) are located in one of the vertical symmetry planes (XY or YZ), and hence they are only allowed to translate horizontally and vertically and rotate in their corresponding planes, which results in three other DOFs ($q_{3},\;q_{4},\;q_{11}$). Each vertex $D_{i}$ (for i=1 to 8) is located in a single vertical symmetry plane (one of two bisectors of YZ and YX), and therefore, it is only allowed to translate horizontally and vertically and rotate in the planes it is located in, which leads to three other DOFs ($q_{5},\;q_{6},\;q_{10}$). Vertices $E_{i}$ and $F_{i}$ (for i=1 to 4) are located in one of two vertical symmetry planes (XY or YZ) and the horizontal symmetry plane XZ, so they are only allowed to translate in the X or Z directions without any rotations. This creates two other DOFs $q_{2}$ and $q_{1}$ at vertices E and F, respectively. Vertices $G_{i}$ are located in the horizontal symmetry plane XZ and in one of the vertical bisector symmetry planes (bisectors of XY and YZ). Therefore, these points cannot rotate in any directions and are only allowed to translate horizontally in the bisector of the noted planes, which leads to the final DOF $q_{9}$. Overall, the system has a total number of 11 DOFs (Figure 2b).

In this study, two well-known elastic beam theories, namely, Euler–Bernoulli and Timoshenko beam theories, have been implemented for analytical analysis of the re-entrant unit cell. As the beam theories and material models used for the analytical analysis are linear, the overall deformation of each DOF in the system can be considered as the superposition of separate deformations caused by applying individual loads at each DOF. Therefore, the superposition principle could be implemented for obtaining the system of equations of the unit cell. In this approach, each DOF displaces autonomously from other DOFs, and when a DOF is displaced, the other DOFs are kept fixed. By solving the equilibrium equations for the vertices of the unit cell, the resultant forces are obtained. By considering this method, the system of equilibrium equations for the structure could be written as {Q}=[K]{q}. In this equation, {Q} is the force vector comprising the external forces acting on the DOFs, [K] is the stiffness matrix of the system, and the {q} is the displacement vector.

Because the derivation of the stiffness matrix elements is based on beam theories, the general deformation of a cantilever beam (Figure 3) can be considered as the resultant of four distinct deformations at the free end of the beam, which are as follows:
(a)a lateral displacement without any rotation, v;(b)a flexural rotation without any lateral displacement, θ;(c)a longitudinal elongation or contraction, u;(d)an axial twist, φ.

The corresponding forces and moments required to create each of the pure above-mentioned deformations are demonstrated in Figure 4 for the Euler–Bernoulli and Timoshenko beam theories. Due to frequent repetition of some terms demonstrated in Figure 4 throughout the whole manuscript, the terms $S_{i}$, $W_{i}$, $T_{i}$, $V_{i}$, and $U_{i}$ are used, which are different for Euler–Bernoulli and Timoshenko beam theories and are listed in Table 1.

To explain the general procedure for deriving the stiffness matrix elements, we could consider a strut PQ with an arbitrary orientation with respect to the global X, Y, and Z coordinate system (aligned with a unit vector $u_{PQ}$), which is deformed by a specific DOF $q_{i}$ at its free end. The schematic representation of such a strut and its deformation due to the DOF is illustrated in Figure 5. As shown in the figure, DOF $q_{i}$ causes a longitudinal and a transverse deformation in the strut PQ. The vector of longitudinal deformation of strut PQ can be obtained by projecting the vector $q_{i}$ on the strut PQ unit vector, i.e., $\left(q_{i}\cdot u_{PQ}\right)u_{PQ}$. Moreover, the vector of transverse deformation of strut PQ can be obtained as $\left(q_{i}-\left(q_{i}\cdot u_{PQ}\right)u_{PQ}\right)$. By multiplying the longitudinal deformation vector of strut PQ by the resultant force value of PQ (i.e., $S_{i}$), the resultant longitudinal force vector for strut PQ can be obtained as $\left(\left(q_{i}\cdot u_{PQ}\right)u_{PQ}\right)S_{i}$, and by multiplying the transverse deformation vector of strut PQ by the resultant transverse force value of PQ (i.e., $T_{i}$), the resultant transverse force vector for this strut can be obtained as $\left(q_{i}-\left(q_{i}\cdot u_{PQ}\right)u_{PQ}\right)T_{i}$. The reaction moment vector, created by the transverse deformation of strut PQ can be found by the cross-product of the unit vector of PQ (i.e., $u_{PQ}$) and the resultant moment vector of the beam (i.e.,$\left(q_{i}-\left(q_{i}\cdot u_{PQ}\right)u_{PQ}\right)V_{i}$), which yields $-\left(u_{PQ}\times \left(q_{i}-\left(q_{i}\cdot u_{PQ}\right)u_{PQ}\right)\right)V_{i}$. The values of $S_{i}$, $T_{i}$, and $V_{i}$ for Timoshenko and Euler–Bernoulli beam theories are listed in Table 1. It is worth noting that the procedure of obtaining the forces and moments for a lateral rotational DOF is very similar to the above-mentioned procedure, and it could be easily performed by replacing the terms $S_{i}$, $T_{i}$, and $V_{i}$ with the terms $W_{i}$, $V_{i}$, and $U_{i}$, respectively.

In the following, the detailed procedure of derivation of the stiffness matrix elements for three distinct DOFs ($q_{1}$, $q_{2}$, and $q_{10}$) are presented. The derivation procedure of other DOFs is similar to the ones presented in the paper, but they are provided in Appendix A.
(a)First DOF: $q_{1}=1$

Here, the elements of the first column of the stiffness matrix are derived by applying the unit displacement $q_{1}=1$ and by keeping all the other DOFs fixed (i.e., = 0). This deformation type displaces point $F_{1}$ horizontally in the Z direction. This deformation only causes elongation in strut EF and it does not affect the other struts. The vector of elongation of strut EF can be obtained by projecting the vector $q_{1}$ on strut EF, i.e., $\left(\vec{q_{1}}\cdot \frac{\vec{E_{1}F_{1}}}{\left|\vec{E_{1}F_{1}}\right|}\right)\frac{\vec{E_{1}F_{1}}}{\left|\vec{E_{1}F_{1}}\right|}\;$ or in a simpler notation$\;\left(q_{1}\cdot u_{E_{1}F_{1}}\right)u_{E_{1}F_{1}}$, where $q_{1}$ is the unit vector parallel to the first DOF, and $u_{E_{1}F_{1}}$ is the unit vector parallel to strut $E_{1}F_{1}$. The same system of notation will be used in the rest of the document. By multiplying the elongation vector of strut EF by the resultant force parameter of strut EF (i.e., $S_{1}$), the resultant force vector can be obtained as $\left(\left(q_{1}\cdot u_{E_{1}F_{1}}\right)u_{E_{1}F_{1}}\right)S_{1}$.

The equilibrium of forces at point $F_{1}$ in the $q_{1}$ direction, i.e., the Z direction, provides the $K_{11}$ element of the stiffness matrix (Figure 6), which is calculated as
(2)$$\sum^{\;}F_{z,F_{1}}=0\;\to \;\frac{Q_{1}}{4}-q_{1}\cdot \left(\left(q_{1}\cdot u_{E_{1}F_{1}}\right)u_{E_{1}F_{1}}\right)S_{1}=0\;\to K_{11}=Q_{1}=4S_{1}$$

It must be noted that the forces demonstrated in Figure 6 are the only forces that must act on the beam in order for it to deform in the way it is shown. The force the beam applies at each vertex is in the opposite direction of what is shown in Figure 6. The same holds true for all the next derivations. The equilibrium of forces at point $E_{1}$ in the $q_{2}$ direction (i.e., in the Z direction) provides the $K_{12}$ element of the stiffness matrix, which is calculated as
(3)$$\sum^{\;}F_{z,E_{1}}=0\;\to \;\frac{Q_{2}}{4}+q_{2}\cdot \left(\left(q_{1}\cdot u_{E_{1}F_{1}}\right)u_{E_{1}F_{1}}\right)S_{1}=0\;\to K_{12}=Q_{2}=-4S_{1}$$

Because the displacement of DOF $q_{1}$ does not have any effect on other DOFs ($q_{3},\;q_{4},\;q_{5},\;q_{6},\;q_{7},\;q_{8},\;q_{9},\;q_{10},\;q_{11})$, the remaining elements of the first column of the stiffness matrix are zero.
(b)Second DOF: $q_{2}=1$

In this subsection, the elements of the second column of the stiffness matrix are derived. Therefore, we set $q_{2}=1$, and the other DOFs are set to zero. This deformation type displaces point E horizontally in the Z direction, which causes strut EF to have a pure contraction and strut ED to have contraction accompanied by transverse displacement. The vector of contraction of strut EF can be obtained by projecting vector $q_{2}$ on strut EF, i.e., $\left(q_{2}\cdot u_{E_{1}F_{1}}\right)u_{E_{1}F_{1}}$. The vector of contraction of strut ED can be obtained by projecting vector $q_{2}$ on strut ED, i.e., $\left(q_{2}\cdot u_{E_{1}D_{1}}\right)u_{E_{1}D_{1}}$, and the vector of transverse displacement of vertex E of strut ED can be calculated as $\left(q_{2}-\left(q_{2}\cdot u_{E_{1}D_{1}}\right)u_{E_{1}D_{1}}\right)$.

By multiplying the contraction vector of strut EF by the resultant force parameter of strut EF (i.e., $S_{1}$), the resultant force vector of strut EF can be obtained as $\left(\left(q_{2}\cdot u_{E_{1}F_{1}}\right)u_{E_{1}F_{1}}\right)S_{1}$. Moreover, by multiplying the elongation vector of strut ED by the resultant force parameter of strut ED (i.e., $S_{2}$), the resultant force vector can be obtained as $\left(\left(q_{2}\cdot u_{E_{1}D_{1}}\right)u_{E_{1}D_{1}}\right)S_{2}$. Finally, by multiplying the transverse deformation vector of strut ED by the resultant transverse force parameter of strut ED (i.e., $T_{2}$), the resultant transverse force vector of this strut can be obtained as $\left(q_{2}-\left(q_{2}\cdot u_{E_{1}D_{1}}\right)u_{E_{1}D_{1}}\right)T_{2}$. It is worth noting that the resultant forces obtained for deformation of struts ED at point E must be multiplied by four because there are four similar struts ED connected to point E. The equilibrium of forces at point E in the $q_{2}$ direction (i.e., the Z direction), Figure 7b, provides the element $K_{22}$ of the stiffness matrix as follows:(4)$$\begin{array}{c} \sum^{\;}F_{z,E_{1}}=0\;\to \;\frac{Q_{2}}{4}-q_{2}\cdot \left(\left(q_{2}\cdot u_{E_{1}F_{1}}\right)u_{E_{1}F_{1}}\right)S_{1}+4\left(q_{2}\cdot \left(\left(q_{2}\cdot u_{E_{1}D_{1}}\right)u_{E_{1}D_{1}}\right)S_{2}\right)-4\left(q_{2}\cdot \left(q_{2}-\left(q_{2}\cdot u_{E_{1}D_{1}}\right)u_{E_{1}D_{1}}\right)T_{2}\right)=0\\ \to K_{22}=Q_{2}=4\left(S_{1}+4\sin^{2}\theta S_{2}+4\cos^{2}\theta T_{2}\right) \end{array}$$

The presence of reaction forces and moments at point D due to $q_{2}$ leads to non-zero values of $K_{25},\;K_{26}$, and $K_{210}$. By projecting the resultant contraction force vector $\left(\left(q_{2}\cdot u_{E_{1}D_{1}}\right)u_{E_{1}D_{1}}\right)S_{2}$ and the transverse deformation force vector $\left(q_{2}-\left(q_{2}\cdot u_{E_{1}D_{1}}\right)u_{E_{1}D_{1}}\right)T_{2}$ of strut ED in the $q_{5}$ and $q_{6}$ directions, and solving the equilibrium of forces at point D in these two directions, the elements $K_{25}$ and $K_{26}$ of the stiffness matrix could be obtained. It is important to note that the force vectors at point D have been multiplied by two (as two ED struts are connected to point D):(5)$$\begin{array}{c} \sum^{\;}F_{y,D_{1}}=0\;\to \;\frac{Q_{5}}{8}+2q_{5}\cdot \left(\left(q_{2}\cdot u_{E_{1}D_{1}}\right)u_{E_{1}D_{1}}\right)S_{2}+2q_{5}\cdot \left(q_{2}-\left(q_{2}\cdot u_{E_{1}D_{1}}\right)u_{E_{1}D_{1}}\right)T_{2}=0\\ \to K_{25}=Q_{5}=4\sqrt{2}\sin2\theta \left(T_{2}-S_{2}\right) \end{array}$$
(6)$$\begin{array}{c} \sum^{\;}F_{q_{6},D_{1}}=0\;\to \;\frac{Q_{6}}{8}+2q_{6}\cdot \left(\left(q_{2}\cdot u_{E_{1}D_{1}}\right)u_{E_{1}D_{1}}\right)S_{2}+2q_{6}\cdot \left(q_{2}-\left(q_{2}\cdot u_{E_{1}D_{1}}\right)u_{E_{1}D_{1}}\right)T_{2}=0\\ \to K_{26}=Q_{6}=4\sin2\theta \left(T_{2}-S_{2}\right)-8\sqrt{2}\sin^{2}\theta S_{2}-8\sqrt{2}\cos^{2}\theta T_{2} \end{array}$$

To obtain the element $K_{210}$ of the stiffness matrix, the equilibrium of moments in the $q_{10}$ direction at point D must be considered. The reaction moment vector, created by transverse deformation of point E of strut ED can be obtained by the cross-product of unit vector of strut ED, $u_{E_{1}D_{1}},\;$and the resultant moment vector of the beam at point D, i.e.,$\;\left(q_{2}-\left(q_{2}\cdot u_{E_{1}D_{1}}\right)u_{E_{1}D_{1}}\right)V_{2}$. This vector must be multiplied by two in the equilibrium equation as two struts ED are connected to point D. Finally, by solving the equilibrium of moments in the $q_{10}$ direction, the element $K_{210}\;$ of the stiffness matrix could be derived:(7)$$\sum^{\;}M_{q_{10},D_{1}}=0\;\to \;\frac{Q_{10}}{8}+2q_{10}\cdot \left(u_{E_{1}D_{1}}\times \left(q_{2}-\left(q_{2}\cdot u_{E_{1}D_{1}}\right)u_{E_{1}D_{1}}\right)\right)V_{2}=0\;\to K_{210}=Q_{10}=-8\cos\theta V_{2}$$
(c)The 10th DOF: $q_{10}=1$

This DOF is a complex deformation in the structure because it applies rotation at vertex D, which is connected to six struts (2 × DC, 2 × DE, DG, and DB). By projecting vector $q_{10}$ on the noted struts, the torsion applied to each of the struts can be found ($q_{10}\cdot u_{C_{1}D_{1}},\;q_{10}\cdot u_{E_{1}D_{1}},\;q_{10}\cdot u_{B_{1}D_{1}},\;q_{10}\cdot u_{G_{1}D_{1}}$). Afterward, by multiplying the torsional parameter of each strut ($W_{2}$ and $W_{3}$) by the projected twist, the torque vectors could be obtained as $\left(\left(q_{10}\cdot u_{C_{1}D_{1}}\right)u_{C_{1}D_{1}}\right)W_{3}$, $\left(\left(q_{10}\cdot u_{E_{1}D_{1}}\right)u_{E_{1}D_{1}}\right)W_{2}$, $\left(\left(q_{10}\cdot u_{B_{1}D_{1}}\right)u_{B_{1}D_{1}}\right)W_{2}$, and $\left(\left(q_{10}\cdot u_{G_{1}D_{1}}\right)u_{G_{1}D_{1}}\right)W_{3}$. The extent of flexural rotation of struts DC, DE, DG, and DB can be obtained by subtracting the projected twist vector from vector $q_{10}$, which yields $q_{10}-\left(q_{10}\cdot u_{C_{1}D_{1}}\right)u_{C_{1}D_{1}},\;q_{10}-\left(q_{10}\cdot u_{E_{1}D_{1}}\right)u_{E_{1}D_{1}},\;q_{10}-\left(q_{10}\cdot u_{B_{1}D_{1}}\right)u_{B_{1}D_{1}}$, and $q_{10}-\left(q_{10}\cdot u_{G_{1}D_{1}}\right)u_{G_{1}D_{1}}$, respectively. Afterward, by multiplying the resultant bending parameter of each strut (i.e., $U_{2}$ and $U_{3}$) to the noted terms, the lateral force vectors could be calculated. It is worth noting that the terms related to struts CD and ED must be multiplied by two because two of such struts are connected to point D. Finally, by projecting all the resultant axial and lateral moments in the $q_{10}$ direction and writing the equation of moment equilibrium in this direction, the element $K_{1010}$ can be found:(8)$$\begin{array}{c} \begin{array}{c} \sum^{\;}M_{q_{10},D_{1}}=0\;\to \;\frac{Q_{10}}{8}-2q_{10}\cdot \left(\left(q_{10}\cdot u_{C_{1}D_{1}}\right)u_{C_{1}D_{1}}\right)W_{3}-2q_{10}\cdot \left(q_{10}-\left(q_{10}\cdot u_{C_{1}D_{1}}\right)u_{C_{1}D_{1}}\right)U_{3}-2q_{10}\\ \cdot (\left(q_{10}\cdot u_{E_{1}D_{1}}\right)u_{E_{1}D_{1}})W_{2}-2q_{10}\cdot \left(q_{10}-\left(q_{10}\cdot u_{E_{1}D_{1}}\right)u_{E_{1}D_{1}}\right)U_{2}-q_{10}\cdot \left(\left(q_{10}\cdot u_{B_{1}D_{1}}\right)u_{B_{1}D_{1}}\right)W_{2}-q_{10}\\ \cdot \left(q_{10}-\left(q_{10}\cdot u_{B_{1}D_{1}}\right)u_{B_{1}D_{1}}\right)U_{2}-q_{10}\cdot \left(\left(q_{10}\cdot u_{G_{1}D_{1}}\right)u_{G_{1}D_{1}}\right)W_{3}-q_{10}\cdot \left(q_{10}-\left(q_{10}\cdot u_{G_{1}D_{1}}\right)u_{G_{1}D_{1}}\right)U_{3}=0 \end{array}\\ \begin{array}{cc} \to K_{1010} & =Q_{10}=\;2(\left(3+\cos2\varphi -2\sqrt{2}\sin2\varphi \right)W_{3}+2W_{2}\left(2\sin^{2}\theta +\cos^{2}\theta -2\sqrt{2}\sin2\theta \right)\\ & +2\sqrt{2}\sin2\theta U_{2}-2\cos^{2}\theta U_{2}+\left(2\left(5+\cos2\theta \right)U_{2}+\left(9-\cos2\varphi +2\sqrt{2}\sin2\varphi \right)U_{3}\right)) \end{array} \end{array}$$

The element $K_{1011}$ could be obtained by solving the equilibrium of moments at vertex C. By projecting the reaction torsion and bending moment vectors created by the rotation of vertex D of strut CD in the $q_{10}$ direction and solving the equations of moment equilibrium in the same direction, the element $K_{1011}$ could be calculated as follows:(9)$$\begin{array}{c} \sum^{\;}M_{q_{11},C_{1}}=0\;\to \;\frac{Q_{11}}{8}-2q_{11}\cdot \left(\left(q_{10}\cdot u_{C_{1}D_{1}}\right)u_{C_{1}D_{1}}\right)W_{3}-2q_{11}\cdot \left(q_{10}-\left(q_{10}\cdot u_{C_{1}D_{1}}\right)u_{C_{1}D_{1}}\right)\frac{U_{3}}{2}=0\\ \to K_{1011}=Q_{11}=\;-4\left(2\sqrt{2}\cos^{2}\varphi -\sin2\varphi \right)W_{3}+2\left(\sin2\varphi +2\sqrt{2}\sin^{2}\varphi \right)U_{3} \end{array}$$

#### 2.1.3. Stiffness Matrix Derivation

Finally, using the stiffness matrix elements obtained in Section 2.1.2 and also by considering the loading condition ($Q_{8}=2F$), the stiffness matrix and system of equations can be constructed as
(10)$$\left\{\begin{array}{c} 0\\ 0\\ 0\\ 0\\ 0\\ 0\\ 0\\ 2F\\ 0\\ 0\\ 0 \end{array}\right\}=\left[\begin{array}{ccc} A_{11} & A_{12} & A_{13} \end{array}\right]\left\{\begin{array}{c} q_{1}\\ q_{2}\\ q_{3}\\ q_{4}\\ q_{5}\\ q_{6}\\ q_{7}\\ q_{8}\\ q_{9}\\ q_{10}\\ q_{11} \end{array}\right\}$$
where F is the external load applied on the unit cell. $\left[A_{11}\right]$,$\left[A_{12}\right]$, and $\left[A_{13}\right]$ are the partial matrices included in the overall stiffness matrix [K]. $\left[A_{11}\right]$,$\left[A_{12}\right]$, and $\left[A_{13}\right]$ have been used for easier representation of the overall stiffness matrix and are as follows:


(11)$$\begin{array}{c} A_{11}=\left[\begin{array}{cccc} 4S_{1} & -4S_{1} & 0 & 0\\ -4S_{1} & 4\left(S_{1}+4\sin^{2}\theta S_{2}+4\cos^{2}\theta T_{2}\right) & 0 & 0\\ 0 & 0 & 8\sin^{2}\theta S_{3}+4\left(3+\cos2\varphi \right)T_{3} & 8\sin^{2}\varphi \left(S_{3}-T_{3}\right)\\ 0 & 0 & 8\sin^{2}\varphi \left(S_{3}-T_{3}\right) & 8\sin^{2}\varphi S_{3}+4\left(3+\cos2\varphi \right])T_{3}\\ 0 & \frac{16\cos\varphi \sin\theta L_{3}\left(-S_{2}+T_{2}\right)}{L_{2}} & 8\sin^{2}\varphi S_{3}+4\left(3+\cos2\varphi \right)T_{3} & 8\sin^{2}\varphi \left(S_{3}-T_{3}\right)\\ 0 & -\frac{8\sqrt{2}\left(\cos\varphi \sin\theta L_{3}\left(S_{2}-T_{2}\right)+L_{2}\left(\sin^{2}\theta S_{2}+\cos^{2}\theta T_{2}\right)\right)}{L_{2}} & 4\sqrt{2}\sin\varphi \left(\sqrt{2}\cos\varphi +\sin\varphi \right)\left(S_{3}-T_{3}\right) & 4\sqrt{2}\left(\sin\varphi \left(\sqrt{2}\cos\varphi +\sin\varphi \right)S_{3}-\left(-2+\sqrt{2}\cos\varphi \sin\varphi +\sin^{2}\varphi \right)T_{3}\right)\\ 0 & 0 & 0 & 0\\ 0 & 0 & 0 & 0\\ 0 & 0 & 0 & 0\\ 0 & -\frac{8\sqrt{2}\cos\varphi L_{3}V_{2}}{L_{2}} & -8\left(\sqrt{2}\cos\varphi +\sin\varphi \right)V_{3} & 8\sin\varphi V_{3}\\ 0 & 0 & -8\sqrt{2}\sin\varphi V_{3} & 8\sqrt{2}\sin\varphi V_{3} \end{array}\right]\\ A_{12}=\left[\begin{array}{cccc} 0 & 0 & 0 & 0\\ \frac{16\cos\varphi \sin\theta L_{3}\left(-S_{2}+T_{2}\right)}{L_{2}} & -\frac{8\sqrt{2}\left(\cos\varphi \sin\theta L_{3}\left(S_{2}-T_{2}\right)+L_{2}\left(\sin^{2}\theta S_{2}+\cos^{2}\theta T_{2}\right)\right)}{L_{2}} & 0 & 0\\ 8\sin^{2}\varphi S_{3}+4\left(3+\cos2\varphi \right)T_{3} & 4\sin\varphi \left(2\cos\varphi +\sqrt{2}\sin\varphi \right)\left(S_{3}-T_{3}\right) & 0 & 0\\ 8\sin^{2}\varphi \left(S_{3}-T_{3}\right) & 4\sin\varphi \left(2\cos\varphi +\sqrt{2}\sin\varphi \right)S_{3}-4\left(-2\sqrt{2}+\sqrt{2}\sin^{2}\varphi +\sin2\varphi \right)T_{3} & 0 & 0\\ 4\left(2\sin^{2}\theta S_{2}+2S_{3}+\frac{4\cos^{2}\varphi L_{3}^{2}\left(S_{2}-T_{2}\right)}{L_{2}^{2}}+5T_{2}+\cos2\theta T_{2}+4T_{3}\right) & \frac{16\sqrt{2}\cos\varphi \sin\theta L_{3}\left(S_{2}-T_{2}\right)}{L_{2}}+\frac{8\sqrt{2}\cos^{2}\varphi L_{3}^{2}\left(S_{2}-T_{2}\right)}{L_{2}^{2}}+4\sin\varphi \left(4\cos\varphi +\sqrt{2}\sin\varphi \right)\left(S_{3}-T_{3}\right) & 8\left(\sin^{2}\theta S_{2}+\cos^{2}\theta T_{2}\right) & 0\\ \frac{16\sqrt{2}\cos\varphi \sin\theta L_{3}\left(S_{2}-T_{2}\right)}{L_{2}}+\frac{8\sqrt{2}\cos^{2}\varphi L_{3}^{2}\left(S_{2}-T_{2}\right)}{L_{2}^{2}}+4\sin\varphi \left(4\cos\varphi +\sqrt{2}\sin\varphi \right)\left(S_{3}-T_{3}\right) & 2\left(4\sin^{2}\theta S_{2}+\left(5-\cos2\varphi +2\sqrt{2}\sin2\varphi \right)S_{3}+\frac{8\cos\varphi \sin\theta L_{3}\left(S_{2}-T_{2}\right)}{L_{2}}+\frac{12\cos^{2}\varphi L_{3}^{2}\left(S_{2}-T_{2}\right)}{L_{2}^{2}}+10T_{2}+2\cos2\theta T_{2}+7T_{3}+\cos2\varphi T_{3}-2\sqrt{2}\sin2\varphi T_{3}\right) & \frac{8\sqrt{2}\cos\varphi \sin\theta L_{3}\left(S_{2}-T_{2}\right)}{L_{2}} & 0\\ 8\left(\sin^{2}\theta S_{2}+\cos^{2}\theta T_{2}\right) & \frac{8\sqrt{2}\cos\varphi \sin\theta L_{3}\left(S_{2}-T_{2}\right)}{L_{2}} & 2\left(S_{1}+4\sin^{2}\theta S_{2}+4\cos^{2}\theta T_{2}\right) & -2S_{1}\\ 0 & 0 & -2S_{1} & 2S_{1}\\ 8\cos\varphi \sin\varphi \left(-S_{3}+T_{3}\right) & -8\left(\sin^{2}\varphi S_{3}+\cos^{2}\varphi T_{3}\right) & 0 & 0\\ -\frac{8\sqrt{2}\left(\sin\theta L_{2}+2\cos\varphi L_{3}\right)V_{2}}{L_{2}}-8\left(\sqrt{2}\cos\varphi +2\sin\varphi \right)V_{3} & 8\left(\left(\sin\theta +\frac{2\cos\varphi L_{3}}{L_{2}}\right)V_{2}+\left(\cos\varphi +\sqrt{2}\sin\varphi \right)V_{3}\right) & -\frac{8\sqrt{2}\cos\varphi L_{3}V_{2}}{L_{2}} & 0\\ -8\sqrt{2}\sin\varphi V_{3} & 8\sin\varphi V_{3} & 0 & 0 \end{array}\right]\\ \mathrm{and}\;\;\;A_{13}=\left[\begin{array}{ccc} 0 & 0 & 0\\ 0 & -\frac{8\sqrt{2}\cos\varphi L_{3}V_{2}}{L_{2}} & 0\\ 0 & -8\left(\sqrt{2}\cos\varphi +\sin\varphi \right)V_{3} & -8\sqrt{2}\sin\varphi V_{3}\\ 0 & 8\sin\varphi V_{3} & 8\sqrt{2}\sin\varphi V_{3}\\ 8\cos\varphi \sin\varphi \left(-S_{3}+T_{3}\right) & -\frac{8\sqrt{2}\left(\sin\theta L_{2}+2\cos\varphi L_{3}\right)V_{2}}{L_{2}}-8\left(\sqrt{2}\cos\varphi +2\sin\varphi \right)V_{3} & -8\sqrt{2}\sin\varphi V_{3}\\ -8\left(\sin^{2}\varphi S_{3}+\cos^{2}\varphi T_{3}\right) & 8\left(\left(\sin\theta +\frac{2\cos\varphi L_{3}}{L_{2}}\right)V_{2}+\left(\cos\varphi +\sqrt{2}\sin\varphi \right)V_{3}\right) & 8\sin\varphi V_{3}\\ 0 & -\frac{8\sqrt{2}\cos\varphi L_{3}V_{2}}{L_{2}} & 0\\ 0 & 0 & 0\\ 8\left(\sin^{2}\varphi S_{3}+\cos^{2}\varphi T_{3}\right) & -8\cos\varphi V_{3} & 0\\ -8\cos\varphi V_{3} & \frac{2\left(\left(3+\cos2\varphi -2\sqrt{2}\sin2\varphi \right)W_{3}L_{2}^{2}+4W_{2}{\left(\sin\theta L_{2}-\cos\varphi L_{3}\right)}^{2}+8\cos\varphi \sin\theta L_{2}L_{3}U_{2}-4\cos^{2}\varphi L_{3}^{2}U_{2}+L_{2}^{2}\left(2\left(5+\cos2\theta \right)U_{2}+\left(9-\cos2\varphi +2\sqrt{2}\sin2\varphi \right)U_{3}\right)\right)}{L_{2}^{2}} & -8\cos\varphi \left(\sqrt{2}\cos\varphi -\sin\varphi \right)W_{3}+4\sin\varphi \left(\cos\varphi +\sqrt{2}\sin\varphi \right)U_{3}\\ 0 & -8\cos\varphi \left(\sqrt{2}\cos\varphi -\sin\varphi \right)W_{3}+4\sin\varphi \left(\cos\varphi +\sqrt{2}\sin\varphi \right)U_{3} & 16\left(\cos^{2}\varphi W_{3}+\sin^{2}\varphi U_{3}\right) \end{array}\right] \end{array}$$


#### 2.1.4. Elastic Properties Relationships

All the unknown displacements and rotations $q_{1}$ to $q_{11}$ can be obtained as functions of external force $F_{UC}$ by inverting the final stiffness matrix $\left[K\right]=\left[\begin{array}{ccc} \;\left[A_{11}\right] & \;\left[A_{12}\right] & \;\left[A_{13}\right] \end{array}\right]$ and multiplying it by the force vector. By having the DOF displacements as functions of the applied external force, unit cell elastic mechanical properties such as elastic modulus, Poisson’s ratio, and yield stress can be obtained. The elastic modulus of the unit cell can be calculated according to the basic definition of elastic modulus as follows:(12)$$E_{UC}=\frac{F_{UC}L_{UC}}{A_{UC}\delta_{UC}}$$
where, $\;F_{UC}$,$\;L_{UC}$,$\;A_{UC}$,$\;\delta_{UC}$, and $E_{UC}$ are the applied external load, unit cell length, unit cell cross-sectional area, unit cell displacement, and elastic modulus of the re-entrant unit cell, respectively. By considering $F_{UC}=F$,$\;\delta_{UC}=2q_{8}$, $L_{UC}=2\left(l_{1}+l_{3}\cos\varphi -l_{2}\sin\theta \right)$, and $A_{UC}=L_{UC}{}^{2}$, the elastic modulus of the re-entrant unit cell can be obtained as
(13)$$E_{UC}=\frac{F}{4\left(l_{1}+l_{3}\cos\varphi -l_{2}\sin\theta \right)q_{8}}$$

Calculation of $q_{8}$ from solving Equation (10) and inserting it in Equation (13) provides the elastic modulus relationship of the re-entrant unit cell. Because the final relationships for mechanical properties of the unit cell are very lengthy, the elastic modulus relationship for a specific case ($\theta =\varphi =0{}^{\circ},\;r_{1}=r_{2}=r_{3}=0.14b$, and $l_{1}=b$) is presented below:(14)$$E_{UC}=\frac{3\pi \left(9r^{6}\left(2\left(1+\sqrt{2}\right)G_{s}+\left(13+9\sqrt{2}\right){Ε}_{s}\right)+2b^{2}r^{4}\left(2\left(5+3\sqrt{2}\right)G_{s}+\left(79+53\sqrt{2}\right){Ε}_{s}\right)\right)}{2b^{2}\left(9r^{4}\left(\left(26+24\sqrt{2}\right)G_{s}+7\left(23+16\sqrt{2}\right){Ε}_{s}\right)+6b^{2}r^{2}\left(\left(58+34\sqrt{2}\right)G_{s}+\left(407+279\sqrt{2}\right){Ε}_{s}\right)+b^{4}\left(8\left(6+5\sqrt{2}\right)G_{s}+\left(568+436\sqrt{2}\right){Ε}_{s}\right)\right)}$$

Poisson’s ratio of the unit cell$\;\nu_{UC}$ can be found by −q1q8. Hence, by calculating $q_{1}$ and $q_{8}$ from Equation (10) and substituting them in this relationship, the Poisson’s ratio formula for the unit cell can be derived. The Poisson’s ratio relationship of the above-mentioned specific case ($\theta =\varphi =0{}^{\circ},\;r_{1}=r_{2}=r_{3}=0.14b$, and $l_{1}=b$) can be obtained as follows:(15)$$\vartheta_{UC}=\frac{12\left(6+5\sqrt{2}\right)b^{4}{Ε}_{s}+6b^{2}r^{2}\left(2G_{s}+\left(5+\sqrt{2}\right){Ε}_{s}\right)-9r^{4}\left(2G_{s}+\left(5+4\sqrt{2}\right){Ε}_{s}\right)}{9r^{4}\left(\left(26+24\sqrt{2}\right)G_{s}+7\left(23+16\sqrt{2}\right){Ε}_{s}\right)+6b^{2}r^{2}\left(\left(58+34\sqrt{2}\right)G_{s}+\left(407+279\sqrt{2}\right){Ε}_{s}\right)+b^{4}\left(8\left(6+5\sqrt{2}\right)G_{s}+\left(568+436\sqrt{2}\right){Ε}_{s}\right)}$$

Evaluating the von Mises stress contours of the FE models with several relative densities showed that vertices B of struts BD are the most critical points (the points with the maximum stress) in the structure. The yielding of the unit cell as a whole occurs when a point in the unit structure reaches the yield stress of the bulk material $\sigma_{ys}$. Therefore, the yield stress of the re-entrant unit cell can be obtained by assuming that the structure yielding occurs when the stress at point B of struts BD reaches the yield stress of the bulk material. Because strut BD is influenced by both bending moment and axial force ($M_{DB},\;F_{DB}$), the maximum stress ($\sigma_{max}$) in point B of this strut could be calculated as follows:(16)$$\sigma_{max}=\frac{M_{DB}.r_{2}}{I_{2}}+\frac{F_{DB}}{A_{2}}$$
where $r_{2}$, $I_{2}$, and $A_{2}$ are the maximum distance of the points on the strut cross section from the neutral axis, area moment of inertia, and cross-sectional area of strut DB, respectively. To obtain the bending moment and axial force in strut DB, all of the lateral deformation and axial elongation of this strut applied by DOFs should be considered. DOFs $\;q_{5}$,$\;q_{6}$, and $\;q_{7}$ apply axial elongation and lateral displacement to ends B and D of this strut. Moreover, DOF $\;q_{10}$ applies twist to point D. Eventually, the axial forces at point B of strut BD can be obtained as follows:(17)$$F_{DB}=\left(q_{5}\sin\theta +q_{6}\cos\theta +q_{7}\sin\theta \right)S_{2}$$

Similarly, the bending moments at point B of strut DB can be obtained as
(18)$$M_{DB}=\left(-q_{5}\cos\theta +q_{7}\cos\theta +q_{6}\sin\theta \right)V_{2}+\left(q_{10}\right)U_{2}$$

By substituting $F_{DB}$ and $M_{DB}$ from Equations (17) and (18) into Equation (16) and inserting $q_{5}$, $q_{6}$, $q_{7}$, and $q_{10}$ from solution of the inverse of Equation (10), the maximum local yield stress in strut DB can be obtained. For the specific case introduced above ($\theta =\varphi =0{}^{\circ},\;r_{1}=r_{2}=r_{3}=0.14b$, and$\;l_{1}=b$), the relationship can be simplified as follows:(19)$$\sigma_{max}=\frac{F\left(2b^{3}r\left(2G_{s}-\left(1+2\sqrt{2}\right)E_{s}\right)+6b^{2}r^{2}\left(2\left(12+7\sqrt{2}\right)G_{s}+31\left(4+3\sqrt{2}\right)E_{s}\right)-3br^{3}\left(2G_{s}+\left(5+4\sqrt{2}\right)E_{s}\right)+18r^{4}\left(2G_{s}+\left(5+4\sqrt{2}\right)E_{s}\right)+b^{4}\left(8\left(6+5\sqrt{2}\right)G_{s}+\left(664+516\sqrt{2}\right)E_{s}\right)\right)}{36b\pi r^{5}\left(2\left(1+\sqrt{2}\right)G_{s}+\left(13+9\sqrt{2}\right)E_{s}\right)+8b^{3}\pi r^{3}\left(2\left(5+3\sqrt{2}\right)G_{s}+\left(79+53\sqrt{2}\right)E_{s}\right)}$$

Because the applied stress equal to the structure yield stress of the re-entrant unit cell $\sigma_{y,UC}$ causes the maximum local yield stress $\sigma_{max}$ at point B, reaching the bulk material yield stress $\sigma_{y,s}$, the normalized yield stress of the unit cell could be obtained by a cross-multiply of structure mean stress $\sigma_{UC}=\frac{F_{UC}}{A_{UC}}$ and $\sigma_{max}$ on the one hand and the relevant yield stresses on the other hand, which is
(20)$$\frac{\sigma_{y,UC}}{\sigma_{y,s}}=\frac{F_{UC}}{A_{UC}.\sigma_{max}}=\frac{F}{4{\left(l_{1}+l_{3}\cos\varphi -l_{2}\sin\theta \right)}^{2}.\sigma_{Max}}$$

Because the idealized 3D re-entrant structure includes vertical struts (strut AB), the unit cell can be vulnerable to buckling under compressive loading. Strut AB in the unit cell structure can be considered as a column under a pure compressive load, and the critical elastic buckling force of the column can be obtained using the classic Euler’s buckling theory as follows:(21)$$P_{critical}=\frac{\pi^{2}E_{s}I_{1}}{{\left(Kl_{1}\right)}^{2}}$$

By considering strut AB as a clamped column based on the periodic boundary conditions, the theoretical K value of the column will be equal to 1 and the critical load of an idealized 3D re-entrant unit cell under compressive load F can be obtained as
(22)$$P_{critical}=\frac{\pi^{2}E_{s}I_{1}}{l_{1}{}^{2}}$$

Because the total load applied to a single unit cell and to strut AB is identical (load F), by considering the $4{\left(l_{1}+l_{3}\cos\varphi -l_{2}\sin\theta \right)}^{2}$ as the cross-sectional area of the unit cell, the critical buckling stress could be obtained as follows:(23)$$\sigma_{critical}=\frac{\pi^{2}E_{s}I_{1}}{4{\left(l_{1}+l_{3}\cos\varphi -l_{2}\sin\theta \right)}^{2}l_{1}{}^{2}}$$

#### 2.1.5. Effective Lengths

One of the important considerations in deriving the analytical relationships of mechanical properties of structures is that the struts are connected rigidly to each other at the vertices of structures. Ideally, the struts are connected to each other from their very endpoints (vertices demonstrated in Figure 2a), and they do not interact with each other from any other point. However, in practice, when several struts approach a vertex, as the cross-section area of the struts is non-zero, some volumes overlap with each other at the vertices, which in turn decreases the effective lengths of struts participating in the deformation of the structure. This phenomenon becomes of much more significance in auxetic lattices because they have very acute angles at the vertices (see, for example, Figure 2 and Figure 8).

In this study, the minimum distances between the joints were considered as the effective length of the struts. This definition for effective length was in accordance with what was observed in the stress and strain distribution of the structure obtained from the finite element (FE) results. In the following, the analytical derivation of effective lengths has been presented.

The effective length $\;l_{1,eff}$ can be calculated by subtracting the intersection length between Type-I and Type-II struts from $\;l_{1}$ as follows:(24)$$l_{1,eff}=l_{1}-\frac{r_{2}}{\cos\theta }-r_{1}\tan\theta$$

The effective length of Type-II strut is calculated by measuring the distance between the intersection of two Type-II struts at point D (struts DB and DE) and the intersection of Type-I and Type-II struts at point B (struts AB and BD), and it can be calculated by subtracting the intersecting lengths from $\;l_{2}$ as
(25)l2,eff=l2−r2tanα22−r2tanθ−r1cosθ
where $\alpha_{2}$ is the angle between two Type-II struts.

Similarly, the overlapping of Type-III strut occurs in the intersection of Type-II and Type-III struts at point D (struts DC and DG) and the intersection of two Type-II struts at point C. The effective length can then be calculated by subtracting the intersection lengths from $\;l_{3}$ as
(26)$$l_{3,eff}=l_{3}-\frac{r_{2}}{\sin\alpha_{1}}-\frac{r_{3}}{\tan\alpha_{1}}-r_{3}\tan\varphi$$
where $\alpha_{1}$ is the angle between Type-II and Type-III struts.

By substituting these effective lengths into the stiffness matrix and solving the system of Equation (10), analytical relationships for mechanical properties of the 3D re-entrant unit cell can be obtained.

It is worth noting that the analytical relationships for angles $\alpha_{1}$ and $\alpha_{2}$ can be obtained as follows:(27)$$\alpha_{1}=arccos\left(\frac{2l_{3}\cos^{2}\varphi +\sqrt{2}l_{3}\cos\varphi \sin\varphi +\sqrt{2}l_{2}\sin\varphi \sin\theta }{2l_{2}}\right)$$
(28)$$\alpha_{2}=arccos\left(\frac{l_{3}\cos\varphi \left(l_{3}\cos\varphi +2l_{2}\sin\theta \right)}{l_{2}{}^{2}}\right)$$

### 2.2. Experimental Tests

A Quantum Generous 3D printer (Persia Company, Iran) with poly-lactic acid (PLA) filaments was used for manufacturing the test specimens. Because the mechanical properties of PLA could be different after the printing process as compared to its properties in the original filament form, in order to obtain the mechanical properties of the bulk material, three types of cylindrical specimens were manufactured and tested under compressive loading. The cylinders were printed in the horizontal (0°), vertical (90°), and inclined (45°) orientations with respect to the build plate direction. Four specimens were manufactured for each cylinder type (12 specimens in total). All the cylindrical specimens had a nominal length of 20 mm and a diameter of 15 mm.

Three sets of re-entrant unit cell specimens with five different relative densities μ=0.1, 0.15, 0.2, 0.25, and 0.3 were manufactured (15 specimens in total), see Appendix A. The dimensions of all the re-entrant unit cell specimens were 7.71 × 7.71 × 7.71 cm^3^. All the specimens (unit cells and cylinders) were manufactured with a layer thickness of 100 μm and infill density of 100%. Because the forces required for collapsing the manufactured specimens were in the range of 750 N to 3700 N, to avoid the inaccuracies caused by the deformation of the test machine itself, all the tests were performed with a large compression test machine (150 kN) for which the deformation in the machine was negligible. More information on the effect of deformation of the mechanical test bench can be found in [59]. The static compression tests were performed using an STM-20 (STANTAM, Iran) mechanical testing bench with a 20 kN load cell (Figure 8). The displacement rate was set to 0.1 mm/min. The normalized elastic modulus and normalized yield stress of specimens were obtained from load-displacement graphs, and the Poisson’s ratio of the unit cells were extracted from the side-view images.

### 2.3. Numerical Modeling

Two FE models were developed in ANSYS software (Pennsylvania, United States) to calculate the mechanical properties of the re-entrant lattice structures. Because analytical formulas of mechanical properties of unit cells have been derived based on Euler–Bernoulli and Timoshenko beam theories, in the first FE model type, Timoshenko beam elements (3D Quadratic finite strain beam, i.e., beam element Type 189 in ANSYS) were used for discretizing the struts of the unit cell. Timoshenko beam theory takes into account the transverse shear deformation. Each strut was divided into five beam elements. The struts were considered to be rigidly connected to each other at the vertices. The linear elastic material properties of the PLA obtained from the experimental tests were adopted. The elastic modulus, Poisson’s ratio, and yield stress of the beam elements were set to 1.93 GPa, 0.33, and 69 MPa, respectively. Periodic boundary conditions were applied to the FE models. The lowermost nodes of the unit cell at point A_2_
Figure 2a) were fully constrained in all the transitional and rotational directions. The top node of the unit cell at point A_1_ (Figure 2a) was displaced downward, but they were not also allowed to have any horizontal displacement or rotation in any direction (Appendix A).

The second FE model type was developed using volumetric elements. This FE model is expected to better take into account the overlapping effect at the joints on the overall mechanical properties of the structure. Five 3D models of the re-entrant unit cell with geometries almost identical to the geometry of the manufactured specimens (relative densities of 0.1, 0.15, 0.2, 0.25, and 0.3) were constructed. An adaptive fine volumetric mesh was implemented for discretizing the models using tetrahedral 10-node elements (SOLID187 elements in ANSYS) to have a fine mesh at the vertices and efficient larger mesh at the middle of the struts. The finest and most accurate options in ANSYS adaptive meshing, i.e., fine resolution (resolution = 6), slow transition, and fine span angle were chosen. The minimum and maximum element sizes for the lowest-density structure (μ=0.1) were 0.3912 mm and 1.78 mm, respectively. The minimum and maximum element sizes for the highest-density structure (μ=0.3) were 0.5786 mm and 2.45 mm, respectively. The number of elements in the lowest-density and the highest-density structures was 470,701 and 437,269, respectively. Similar to the beam elements, the elastic modulus and Poisson’s ratio of 1.93 GPa and 0.33, respectively, were considered for the bulk material properties. To apply the loading and boundary condition, the lowermost face of the model (lower cross-sectional face of vertex of the strut in Figure 2) was fixed in all the directions and the top face of the model at the point was displaced downward. The material properties assigned to the volumetric elements were the same as those used in the first FE model.

In order to calculate the elastic modulus of the FE models, the sum of reaction force(s) at the lowermost node(s), $F_{UC}$, due to the vertical displacement of the top node(s), $\delta_{UC}$, was measured and substituted in $\;E_{UC}=\frac{F_{UC}L_{UC}}{A_{UC}\delta_{UC}}$, where $A_{UC}$ is the cross-sectional area of the FE model in the plane normal to the loading direction, and $L_{UC}$ is the length of the FE structure in the direction parallel to the loading direction.

For obtaining the Poisson’s ratio, the elongation of the structure in the lateral direction (X or Z direction) was divided by the contraction of the structure in the Y direction (loading direction).

Eventually, for obtaining the numerical normalized yield stress, the maximum stresses (von Mises stress) in strut DB of the unit cell ($\sigma_{max}$) was extracted and inserted into the equation $\frac{F_{UC}}{A_{UC}.\sigma_{max}}$.

## 3. Results

In this section, all the obtained analytical, numerical, and experimental results are presented and compared to each other. To have a better understanding of the main factors influencing the overall performance of the auxetic lattice structure, the effect of each main geometrical parameter of the structure such as relative density, angle θ, and angle φ are studied individually. Moreover, a lightened unit cell, obtained by eliminating the struts $l_{3}$ from the general 3D re-entrant unit cell, is introduced and its mechanical properties are presented. More details on the results can be found in Appendix A.

### 3.1. Effect of Relative Density

According to analytical relationships of mechanical properties of the re-entrant unit cell obtained in Section 2.1, elastic modulus, Poisson’s ratio, and yield stress of the structure depend on seven geometrical main parameters, namely, $r_{1},\;r_{2},\;r_{3},l_{1},\;l_{2}\;or\;l_{3},\;\theta ,$ and φ. To validate the analytical relationships, the mechanical properties of a specific case has been considered in which all the struts have the same radii ($r_{1}=r_{2}=r_{3}$), angles θ and φ are considered to be equal to 22.5°, and $l_{1}=b$ where b is shown in Figure 2b. It is worth noting that all the additively manufactured specimens have also been constructed according to the above-mentioned geometrical parameters. Effects of variations in the relative density of 3D re-entrant unit cell with respect to angles θ and φ is shown in Appendix A.

The analytical (based on Euler–Bernoulli and Timoshenko beam theories), numerical (based on beam and volumetric elements), and experimental results of this case are presented and compared in Figure 9. The main take-out of this figure is that both the Euler–Bernoulli and Timoshenko analytical models after consideration of the effective length, the numerical models based on volumetric elements, and the experimental data points have good agreement with respect to each other. On the other hand, the results of both the Euler–Bernoulli and Timoshenko analytical models without consideration of the effective length and the numerical model based on beam elements (rather than volumetric elements) are close to each other but quite far from the experimental data points. As it can be seen in Figure 9, the analytical models considering the overlapping effect and the numerical models made from volumetric elements have higher levels of stiffness and yield stress, and lower levels of Poisson’s ratio in comparison with the analytical models without considering the overlapping effect and their equivalent numerical models, which are based on beam elements.

As for the case of analytical relationships that neglect the overlapping effect at the vertices, it is evident that as compared to the analytical results based on the Euler–Bernoulli beam theory, the analytical results based on Timoshenko beam theory have much better agreement with the beam-based FE model results for all relative densities (Figure 9). By increasing the relative density, the difference between the numerical and analytical results for both beam theories increase, and the maximum difference between the numerical and analytical results (at a relative density of μ=0.4) for normalized elastic modulus are 7.91% and 0.62% for Euler–Bernoulli and Timoshenko beam theories, respectively. As for the Poisson’s ratio, the maximum difference between the beam-based FE model and the analytical models based on Euler–Bernoulli and Timoshenko beam theories are 14.84% and 0.91%, respectively. It is worth noting that the maximum negative Poisson’s ratio for this case is very high and equal to −0.7791. These numerical/analytical differences for normalized yield stress are 13.76% and 5.32% for Euler–Bernoulli and Timoshenko beam theories, respectively.

The maximum differences between the elastic modulus results of numerical models made from volumetric elements and the analytical models, which consider the overlapping effect, are 54.97% and 45.75% for Euler–Bernoulli and Timoshenko theories, respectively (Figure 9b). These differences for Poisson’s ratio are 85.7% and 88.24% for Euler–Bernoulli and Timoshenko beam theories, respectively (Figure 9a). Finally, the noted differences for normalized yield stress are 32.31% and 6.85% for Euler–Bernoulli and Timoshenko beam theories, respectively (Figure 9c).

The mean values of the mechanical properties of the PLA filament obtained from compression tests on cylindrical specimens are presented in Table 2. All the auxetic specimens failed from the struts that were expected to have the maximum local stress in the unit cell (strut DB; Figure 10 shows the specimens from the top view). The maximum differences between the experimental results and volumetric element numerical results for normalized elastic modulus, Poisson’s ratio, and normalized yield stress are 22.79%, 16.58%, and 15.59%, respectively. The maximum differences between the experimental results and Timoshenko analytical (by considering the overlapping effect) results for normalized elastic modulus, Poisson’s ratio, and normalized yield stress are 28.52%, 25.95%, and 18.14%, respectively.

Figure 11a shows the von Mises stress distribution in the unit cell’s struts for relative density of μ=0.2. From this figure, it can be inferred that the maximum von Mises stress occurs in the same struts expected in analytical and numerical results with beam elements (struts BD). The stress distribution can be better visualized in the sliced unit cell in Figure 11b. The important point to note regarding the numerical results of this FE model is that the maximum stress locations are at the central point of vertex B (and experimental test results). This, again, signifies the importance of considering the effective lengths rather than vertex-to-vertex lengths in the analytical models. The stress and strain distribution for all other unit cells with various relative densities are presented in Appendix A.

### 3.2. Effect of Angles θ and φ

Angles θ and φ are the most critical parameters in the 3D re-entrant structure because these parameters play the main role in creating the negative Poisson’s ratio effect (Appendix A). Therefore, analyzing the effect of these parameters could be very beneficial for predicting the most extreme values of the mechanical properties, in particular, the negative Poisson’s ratio. Using 3D surface plot demonstration, Figure 12 presents variations in the mechanical properties of 3D re-entrant unit cell with respect to angles θ and φ each varying in the range of −25° to +25°. Moreover, the special cases of θ=0°, φ=0°, and θ=φ are marked in the 3D plots with solid, dashed, and dashed-dot lines, respectively. All the plots are presented for $r_{1}=r_{2}=r_{3}=0.14b$ and with considering a constant volume for the circumscribed cube of the unit cell. Although one might assume that the mechanical properties of the unit should have symmetry with respect to angle θ, the results shown in Figure 12 demonstrate that the mechanical properties are not symmetrical with respect to θ=0°. Due to the constant volume assumed for circumscribed cube of unit cell, the length $l_{1}$ changes based on angle θ, and $l_{1}$ is shorter for θ<0 than for θ>0. Therefore, the results are not symmetrical with respect to θ=0°, and we have higher levels of normalized elastic modulus and yield stress for θ<0 as compared to θ>0. According to Figure 12a,c, the maximum normalized elastic modulus of 0.002234 occurs at θ=−25° and φ=2°, and the maximum normalized yield stress of 0.008441 occurs at θ=−25° and φ=4.5°. The behavior of Poisson’s ratio with respect to variations of angles θ and φ is relatively different in comparison with elastic modulus and yield stress. As shown in Figure 12b, the maximum positive and negative values of Poisson’s ratio occur at θ=φ=−25° and θ=−φ=25°, respectively. The graph also demonstrates that by proper combination of θ and φ, the structure can be either auxetic or non-auxetic and with the desired extent. Moreover, the graph shows that to have a near-zero Poisson’s ratio, the angle θ needs to remain in the range of 4°<θ<8.5°, while angle φ can take any value from −25° to +25°. To understand better the effect of angles θ and φ on the Poisson’s ratio of re-entrant structure in other relative densities, the variations of Poisson’s ratio for four relative densities (r/b=0.1, r/b=0.2, r/b=0.3, and r/b=0.4) are presented in Figure 13. According to this figure, the range of Poisson’s ratio changes between −0.6 and +0.2 for r/b=0.1. The range of change in Poisson’s ratio becomes much narrower for higher values of relative density. For instance, the minimum and maximum values of Poisson’s ratio are −0.12 and +0.17 for r/b=0.4. The influences of effective lengths on mechanical properties for various values of angles θ and φ are presented in Appendix A.

### 3.3. Effect of Relative Density on the Mechanical Properties of the Special Case of θ=φ=0°

A critical case can be considered for the structure in which angles θ and φ are considered to be zero. The effect of variation in relative density of such structure is plotted in Figure 14. The figure shows that in case of neglecting the overlapping effect, as compared to the analytical results based on Euler–Bernoulli beam theory, the analytical results based on Timoshenko beam theory have better agreement with numerical results of the FE models based on beam elements for all values of relative density. According to Figure 14a, the re-entrant structure does not have a negative Poisson’s ratio, and the value of Poisson’s ratio for all relative densities is positive and varies from 0.127 to 0.074. Considering the overlapping effect decreases the values of Poisson’s ratio, and the minimum value of Poisson’s ratio for this case reaches 0.009158.

### 3.4. Elimination of Strut $l_{3}$

According to the contours of mechanical properties of 3D re-entrant unit cell shown in Figure 13, the angle φ had negligible effect on the sign of Poisson’s ratio in all values of rb. This gives an inspiration to create a new lightened re-entrant unit cell by eliminating strut $l_{3}$ from the general 3D re-entrant unit cell. The lightened unit cell with negative, zero, and positive values of θ are shown in Figure 15. The analytical relationships for this unit cell could be easily obtained by setting $r_{3}=0$ in the general relationships (i.e., Equations (14), (15) and (20)). The resulting mechanical properties of the lightened re-entrant unit cell are presented in Figure 16. The results show that the maximum normalized elastic modulus and yield stress for the lightened unit cell occur at θ = +25° for all values of relative densities. The maximum negative and positive Poisson’s ratio for the lightened unit cell is obtained as −0.6322 and +0.352, respectively. The variations of mechanical properties of the lightened unit cell with respect to relative density in the special condition of θ=0 have been presented with solid lines in Figure 16. Similar to the general unit cell, the lightened structure has positive Poisson’s ratio for all values of relative densities in θ=0, and the value of Poisson’s ratio for this case varies from 0.1688 to 0.09837 (Figure 16b).

Comparison of the general idealized 3D re-entrant unit cell and lightened 3D re-entrant unit cell can provide good insights for designers to decide which design to choose from. Figure 17 shows the mechanical properties contours for general and lightened unit cells in various values of θ and r/b. For quantitative comparison, the comparisons are performed for the mean values of both relative density and r/b, i.e., μ=0.25 and r/b=0.25. As for the case of μ=0.25, the value of normalized elastic modulus in θ=25° were calculated as 0.006095 and 0.009178 for the general re-entrant and lightened structures, respectively. As for the case of r/b=0.25, the values of normalized elastic modulus were obtained as 0.005367 and 0.004184 for the general re-entrant and lightened structures, respectively (Figure 17 and Appendix A). Moreover, the normalized yield strength at μ=0.25 were 0.007666 and 0.01157 for the general re-entrant and lightened structures, respectively (Appendix A), while at r/b=0.25, they were 0.006837 and 0.005808, respectively (Figure 17 and Appendix A). Therefore, based on results, we can conclude that the lightened re-entrant structure has better strength and stiffness at the same relative densities in comparison with the general re-entrant structure. In contrast, the lightened re-entrant unit cell is weaker than the general re-entrant structure at the same ratio of r/b (Figure 17). This comparison could be drawn for Poisson’s ratio as well. At the same relative density of μ=0.25 and θ=25°, Poisson’s ratio of the general and lightened re-entrant structures were obtained as −0.2191 and −0.2673, respectively. These values at the conditions of r/b=0.25 and θ=25° were −0.2318 and −0.3419 for the general and lightened re-entrant structures, respectively (Figure 17 and Appendix A). Therefore, eliminating strut $l_{3}$ always increases the extent of either the positive or negative values of Poisson’s ratio for both the cases of identical relative densities and r/b ratio.

Appendix A presents the effect of effective lengths on mechanical properties of both the general and lightened 3D reentrant structure for a specific value of θ=φ=25°. According to this figure, it is clear that the influence of effective length in both structures is similar, and considering the effective lengths due to overlapping of the struts at the vertices increases the stiffness and strength of the structure. In addition, the values of Poisson’s ratio of both general and lightened structures are decreased due to the higher resistance to deformation by considering the effective lengths. In general, based on Appendix A, it can be concluded that the effective length’s influence on the mechanical properties of the lightened structure as compared to its influence on the general re-entrant unit cell are similar. Additional data and figures for comparison between the general and lightened re-entrant structures can be found in Appendix A.

## 4. Discussion

### 4.1. Why Do Timoshenko Relationships Have Better Accuracy?

In this paper, the relationships based on Timoshenko beam theory have been derived for elastic modulus, Poisson’s ratio, and yield strength of the re-entrant lattice structure. Based on the presented results in all cases of studies and also in the all relative densities of structure, the Timoshenko beam theory presents better and more precise results for mechanical properties of the lattice structure in comparison with the Euler–Bernoulli beam theory. The main reason for the accuracy of the Timoshenko beam theory is that this theory takes into account shear deformation and rotational bending effects, making it suitable for describing the behavior of thick beams. For this reason, the results of the analytical model based on Euler–Bernoulli theory deviates from numerical results in higher values of relative densities, while the Timoshenko beam theory results overlap with the numerical results in all values of relative density. It worth noting that taking into account the shear deformation effect lowers the flexural stiffness of the struts and hence the stiffness of the structure, which yields to larger deflections of the struts under a static load. This leads to higher Poisson’s ratio and lower elastic modulus values for Timoshenko beam theory in comparison with Euler–Bernoulli beam theory.

### 4.2. The Most Influencing Parameters on the Structure’s Poisson’s Ratio

In general, angles θ and φ are the main design parameters in creating a negative Poisson’s ratio effect in the re-entrant structure. According to the results (Figure 12 and Figure 13), the magnitude of θ determines the sign of Poisson’s ratio (positive or negative). Increasing θ leads to more negative values of Poisson’s ratio in the unit cell. It is important to note that even though the value of φ has some effects on the amount of Poisson’s ratio, but it does not have any effects on the sign of Poisson’s ratio. It is worth mentioning that there exists a specific case (for specific values of θ and φ) in the re-entrant structure where the Poisson’s ratio of structure becomes zero.

Considering the effective length in the analytical calculations decreases the effects of θ and φ, which is a representation of what would be experienced in real-life conditions (as also confirmed by numerical models based on volumetric elements and experimental results). More importantly, the effect of variation in φ on Poisson’s ratio is almost eliminated when the effective length is taken into consideration (see Appendix A).

According to stress distribution in the re-entrant structure, struts BD are the most influential elements in determining the sign and value of the transverse deformation of the unit cell. Applied load at point A of the unit cell is directly transferred to point B of strut BD via strut AB and this causes strut BD to have normal tensile stress, as shown in Figure 18. However, strut CD is under compressive normal stress due to the load applied to the structure. This makes strut CD demonstrate inverse behavior and counteract regarding the extent of negative Poisson’s ratio. Therefore, the effect of angle θ on Poisson’s ratio is much greater than that of φ, either effective lengths are considered or not.

The other parameter effective on the structure’s Poisson’s ratio is relative density—the higher the relative density, the lower the negative Poisson’s ratio will be. An ideal model (analytical beam model without consideration of vertex overlapping) has a negative Poisson’s ratio in all values of relative densities. However, in more realistic models, and particularly in higher relative densities, (very small) positive Poisson’s ratios can be observed (Figure 9). This is due to the positive Poisson’s ratio of the bulk material (PLA), which takes more dominance in the structural deformation of the system in higher relative densities.

### 4.3. Differences between General and Lightened 3D Re-Entrant Structures

Based on the normal and bending stress distribution in the struts of the structure shown in Figure 18, it was predictable that eliminating struts CD from the system could contribute to an increase in the value of unit cell Poisson’s ratio, even though it would lead to a decrease in the strength and stiffness of the structure. According to comparison results presented in Figure 17, the stiffness and strength of the lightened structures resulted from eliminating struts CD are lower than the general re-entrant structure in the same values of r/b. However, it must be mentioned that the mechanical properties of the lattice structures and metamaterials are usually compared with each other at similar relative densities. Thus, Appendix A could be a better reference for comparison of the stiffness and strength of the general and lightened re-entrant unit cells. According to these figures, we can conclude that at the same relative densities, the lightened structure could be a better option in comparison with the general structure due to its higher elastic modulus and yield stress. In addition, the results show that the lightened structure could provide higher values of negative or positive Poisson’s ratio either in the same relative density or r/b ratio.

All of the above-mentioned comparisons are based on the assumed condition of θ=φ in the general 3D re-entrant structure. Therefore, it should be emphasized that the general re-entrant structure can present a much wider range of mechanical properties for θ≠φ as compared to lightened unit cell due to its additional parameters φ and $l_{3}$. Moreover, with a proper choice of θ and φ values, the general structure can present ideal modulated mechanical properties with the desired combination of mechanical properties (Poisson’s ratio, elastic modulus, and yield strength). For instance, as it can be seen in Figure 12b, angle φ has a relatively low effect on Poisson’s ratio (θ is the governing parameter here), while it has a huge effect on elastic modulus and yield strength. Therefore, as compared to lightened structure, the general re-entrant unit cell has the advantage of giving the opportunity of designing a structure with a desirable combination of elastic modulus and Poisson’s ratio.

### 4.4. Some Points Regarding Effective Length (Definition, Effect, Discreparancies, etc.)

All the struts in the analytical solution of the structures were considered as beams, and they were considered to be rigidly connected to each other at the vertices. However, in practice, when the struts reach each other at the joints, they overlap with each other and, in fact, the lengths that contribute to the deformation of struts and the structure as a whole are decreased. Generally, there is no accurate definition for the effective length, and in this study, the minimum distance between the two sides of struts that are not in the overlapping regions has been considered as struts’ effective length. This definition has been chosen based on the stress and strain distribution observations in the struts of the volumetric numerical models. In other words, the effective lengths have been defined by the regions of the struts where stress and strain values start to increase abruptly. By considering the effective length, the analytical results give much better agreement with experimental test results which can be very helpful in analyzing the structure in real-life conditions.

### 4.5. Failure of Idealized 3D Re-Entrant Structure

The idealized 3D re-entrant structure could be considered as a bending dominant lattice structure due to its architecture with several sloped struts. Figure 18 shows the difference between the levels of normal and bending stress components in the struts during a compressive loading condition. According to this figure, it is obvious that the bending stress in both contracted and stretched regions of the struts plays the main role and it has higher magnitudes than normal stress. Therefore, the failure of the structure occurs in the struts with the highest bending stress levels (i.e., strut BD), which is also validated by experimental test results as shown in Figure 10. One would argue that other failure scenarios such as buckling of the struts could come into play, especially in lower values of structure’s relative density. The analytical relationship of critical stress for buckling of the idealized 3D re-entrant structure was derived and presented in Section 2.1.4 (Equation (23)). By calculating the values of buckling stress (Appendix A), it is found that the level of buckling stress for the unit cell is several orders of magnitude higher than yield stress (shown in Figure 9c). Hence, the buckling failure is very unlikely to happen in the idealized 3D re-entrant structure, especially in the range of relative densities applicable for fabricating structures by additive manufacturing.

### 4.6. Limitations and Suggestions

Although the exact analytical solutions based on Euler–Bernoulli and Timoshenko beam theories presented in this study were capable of providing acceptable results for mechanical properties of idealized 3D re-entrant unit cells and their corresponding lattice structure, these relationships are only valid for linear elastic region and small deformations of unit cells. Some of the applications of lattice and cellular metamaterials need large deformations and nonlinear behavior of unit cells to present some extraordinary functionalities [60,61]. For this aim, the material model and the theory for the analytical solution need to be considered as nonlinear elastic or hyper elastic to obtain accurate and realistic results. Hence, the nonlinear solution and analytical relationships, which are valid for nonlinear regions, will be a very valuable step that could be taken for further studies.

For some unit cell types, due to complexity of unit cell shape, asymmetry in the unit cell, asymmetry in the loading condition, size of the structure (being constructed by a large number of cells), and nonlinearity in the deformation (post-yielding behavior), it is not possible to derive analytical solution and/or to perform computational calculations using FE models with micro-structural topology cost-effectively. In such instances, the Cauchy continuum technique [62] could be very beneficial because it considers the behavior of each cell similar to the behavior of continuous materials (without pores). Considering the lattice structures as a continuous material, of course, will bring in some errors in the results because it might not consider some microstructural features. Fortunately, we were able to perform both the analytical and numerical analyses on the more accurate micro-structural models due to symmetry in the unit cell, its deformation being in the linear elastic range, and the possibility of considering the deformation of a unit cell as being representative of the deformation of a lattice structure.

## 5. Conclusions

In this paper, a new complex 3D re-entrant structure with negative Poisson’s ratio was studied and analytical relationships for its elastic mechanical properties (elastic modulus, yield strength, and Poisson’s ratio) were derived. Two different FE models, one based on beam elements and the other based on volumetric elements, were implemented for validating the analytical results. Furthermore, five sets of specimens with different relative density values were additively manufactured and tested under compressive loading conditions, and the obtained experimental data were used for the analysis of the behavior of the structure in actual conditions. The results of the analytical models, which were based on effective length, had good agreement with the volumetric numerical model and experimental results. The effect of various parameters on the mechanical properties of the structure was also studied, and the results demonstrated that the angle θ has the highest effect on the sign of the Poisson’s ratio, while the angle φ has the lowest effect on the Poisson’s ratio. However, the struts corresponding to angle φ provide strength and stiffness for the structure. The results also showed that the structure could have zero Poisson’s ratio for a specific range of θ and φ angles.

In the end, a lightened re-entrant structure was introduced, and its results were compared to those of the general re-entrant structure with θ=φ. The results showed that at the same relative densities, the lightened structure could be a better option in comparison with the general structure due to its higher mechanical stiffness and strength. In addition, the results showed that the lightened structure could provide higher values of negative or positive Poisson’s ratio. It should be emphasized that the general re-entrant structure can present a much wider range of mechanical properties for θ≠φ as compared to the lightened unit cell due to its additional geometrical parameters. With a proper choice of θ and φ values, the general re-entrant structure can present ideal modulated mechanical properties with the desired combination of mechanical properties (Poisson’s ratio, elastic modulus, and yield strength). In summary, the general re-entrant unit cell presented in this research has the advantage of giving the opportunity of designing a structure with a desirable combination of elastic modulus and Poisson’s ratio.

The wide range of mechanical properties such as Poisson’s ratio and elastic modulus provided by the idealized 3D re-entrant and lightened structures create valuable possibilities to utilize these structures in lattice meta-implants with more compatible stress and strain distributions. In addition, unlike many 3D re-entrant unit cells, the idealized 3D re-entrant unit cell studied here can effectively be used in lattice structures with graded Poisson’s ratio distribution due to the fact that no strut is shared with adjacent cells. Hence, the results of this study can be efficiently used in designing and manufacturing meta-implants with the graded distribution of Poisson’s ratio to enhance the micromotion at the bone-implant interfaces.

## Figures and Tables

**Figure 1 materials-14-00993-f001:**
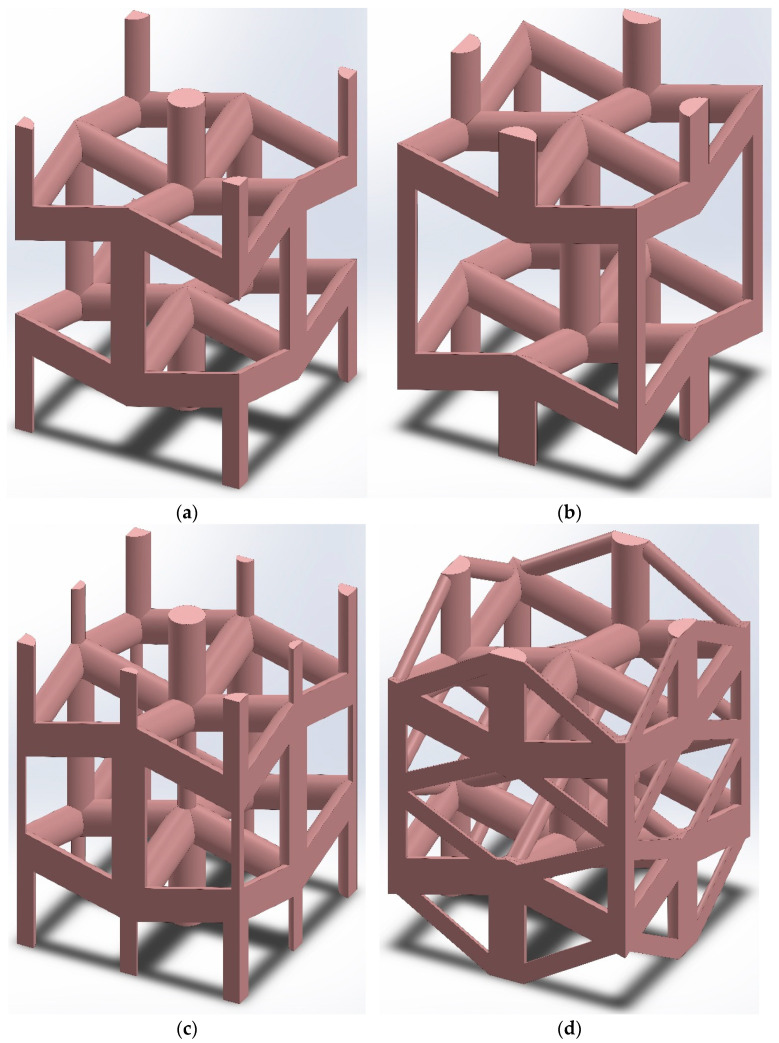
Review of 3D re-entrant unit cells presented by (**a**) Li Yang et al. (2015) [50], (**b**) Xin-Tao Wang et al. (2016) [51], (**c**) Yu Chen et al. (2017) [52], and (**d**) Yingying Xue et al. (2020) [53].

**Figure 2 materials-14-00993-f002:**
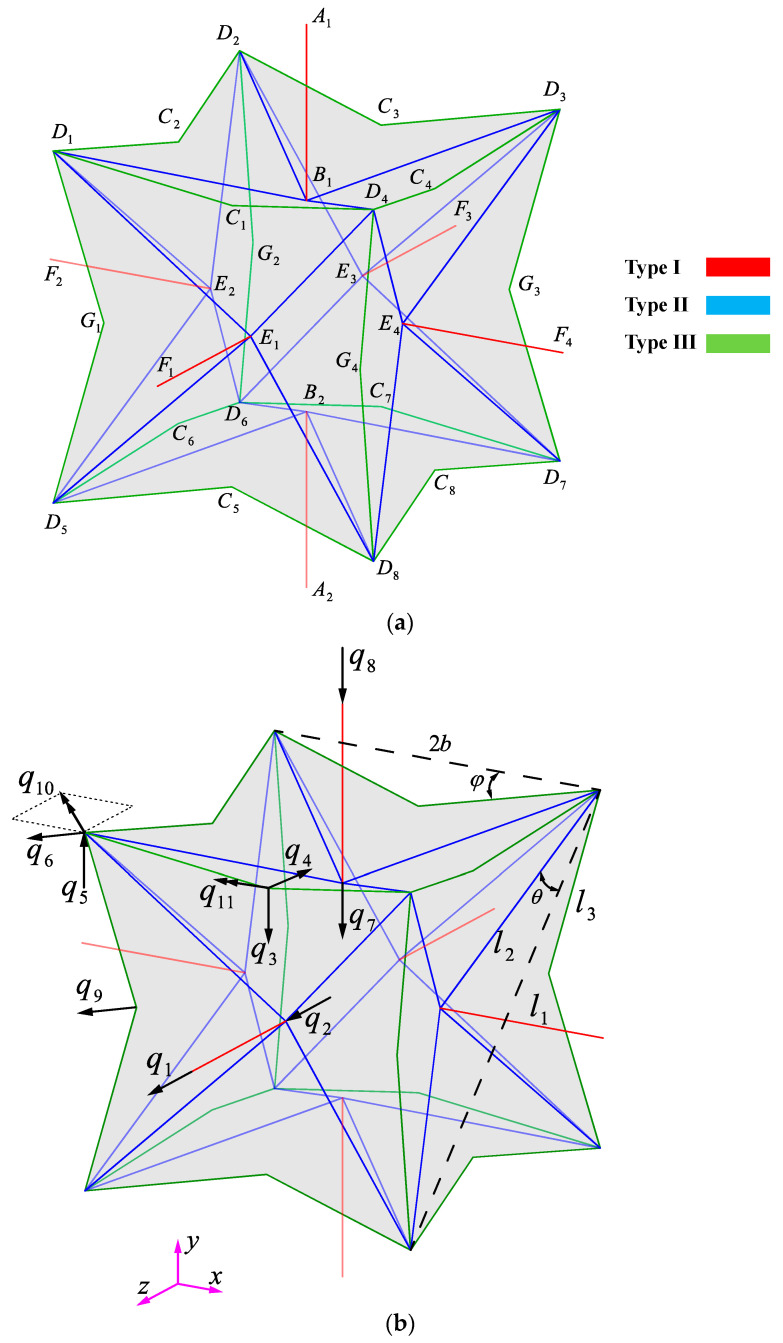
(**a**) An idealized 3D re-entrant unit cell and its strut types, (**b**) degrees of freedom (DOFs) of the unit cell. The double arrows represent the rotational degrees of freedom. More illustrations of the unit cell, its dimensions, and boundary conditions can be found in Appendix A.

**Figure 3 materials-14-00993-f003:**
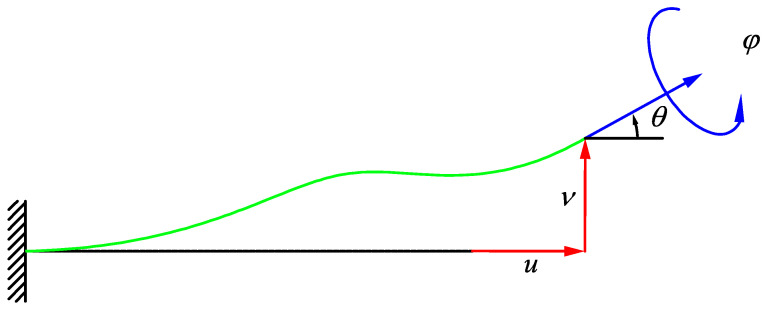
General deformation of a cantilever beam with axial and lateral displacements in addition to flexural and torsional rotations in its free end.

**Figure 4 materials-14-00993-f004:**
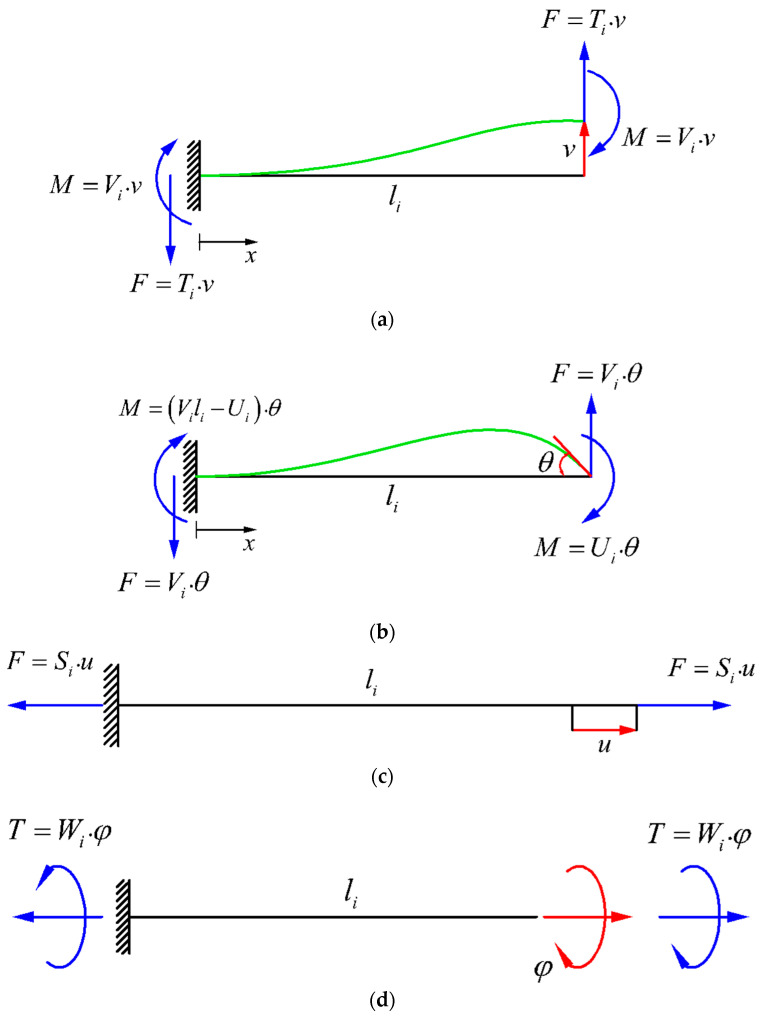
Forces and moments required to cause (**a**) lateral displacement δ with no rotation, (**b**) flexural rotation θ with no lateral displacement, (**c**) pure axial extension, and (**d**) pure twist at the free end of a beam [58].

**Figure 5 materials-14-00993-f005:**
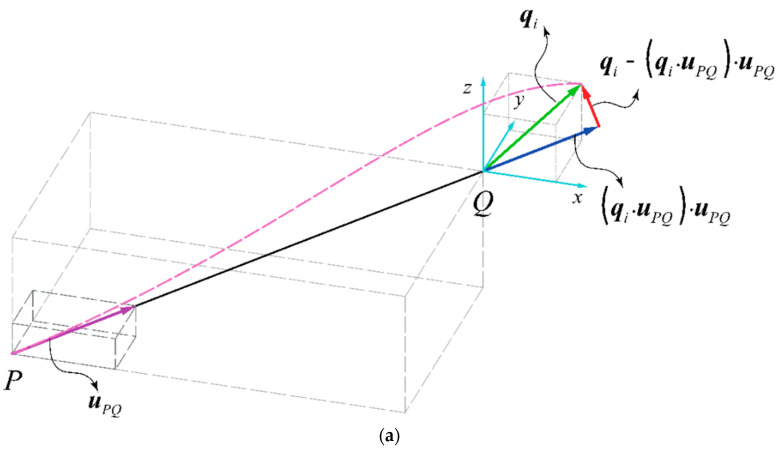
General deformation of a strut due to $q_{i}=1$ (**a**) magnitude of deformations and (**b**) resultant forces and moments.

**Figure 6 materials-14-00993-f006:**
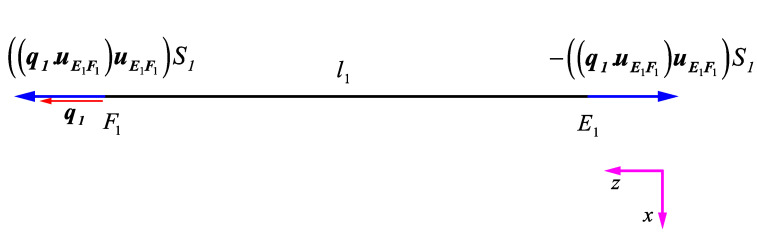
Deformation of strut EF due to
$q_{1}=1$.

**Figure 7 materials-14-00993-f007:**
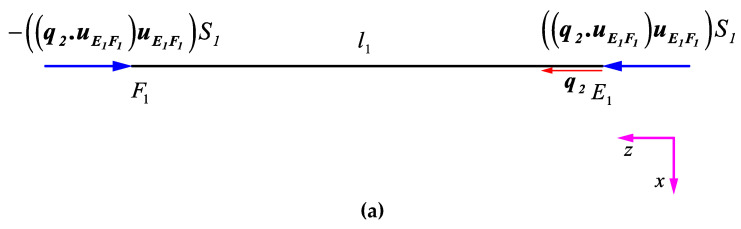
Deformation and loads applied on struts (**a**) EF and (**b**) ED due to $q_{2}=1$.

**Figure 8 materials-14-00993-f008:**
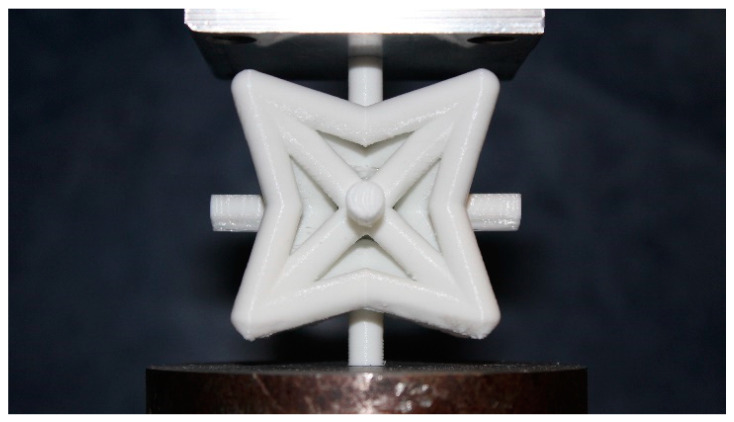
An additively manufactured specimen with μ=0.25 under loading test.

**Figure 9 materials-14-00993-f009:**
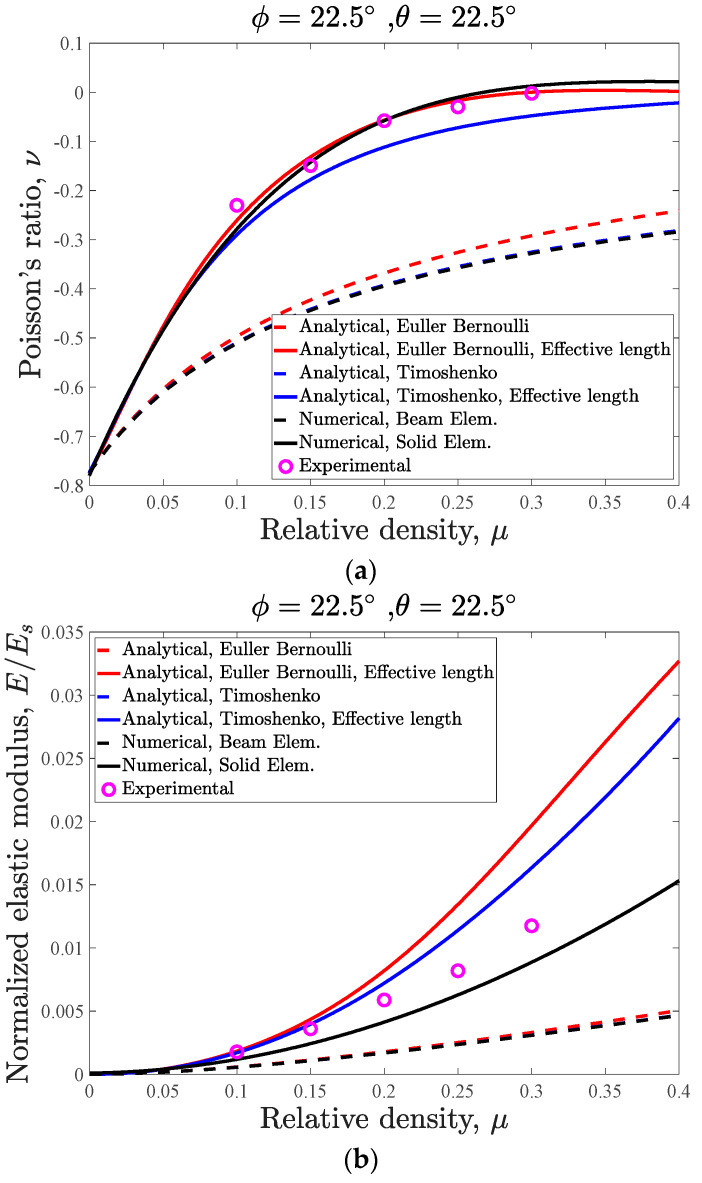
Mechanical properties curves of 3D re-entrant structure with φ=22.5, θ=22.5: (**a**) Poisson’s ratio, (**b**) elastic modulus, and (**c**) yield stress.

**Figure 10 materials-14-00993-f010:**
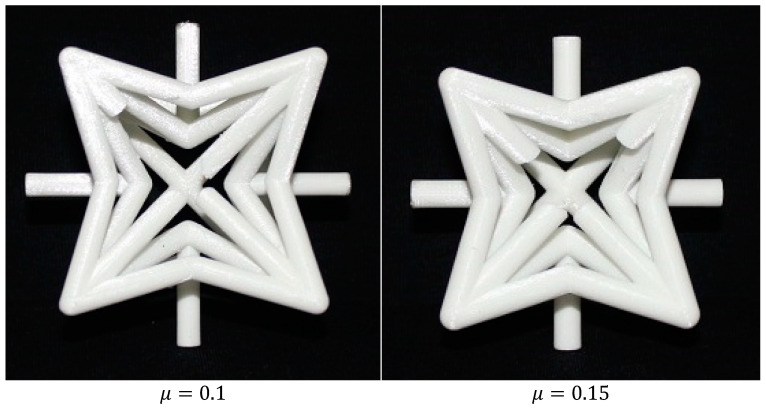
Top view of specimens with different relative densities after the final fracture.

**Figure 11 materials-14-00993-f011:**
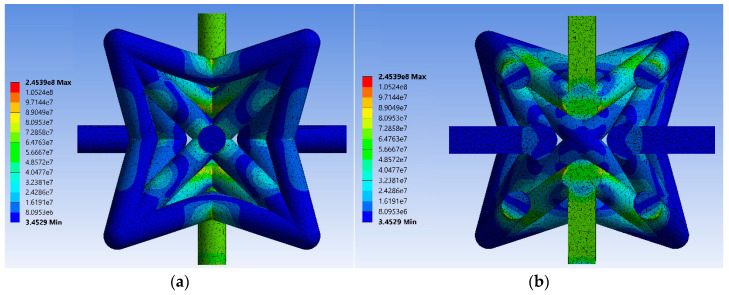
Von Mises equivalent stress for finite element (FE) model constructed by volumetric elements for μ=0.2: (**a**) side view and (**b**) middle-section view of the unit cell. Von-Mises for all other relative densities are shown in Appendix A.

**Figure 12 materials-14-00993-f012:**
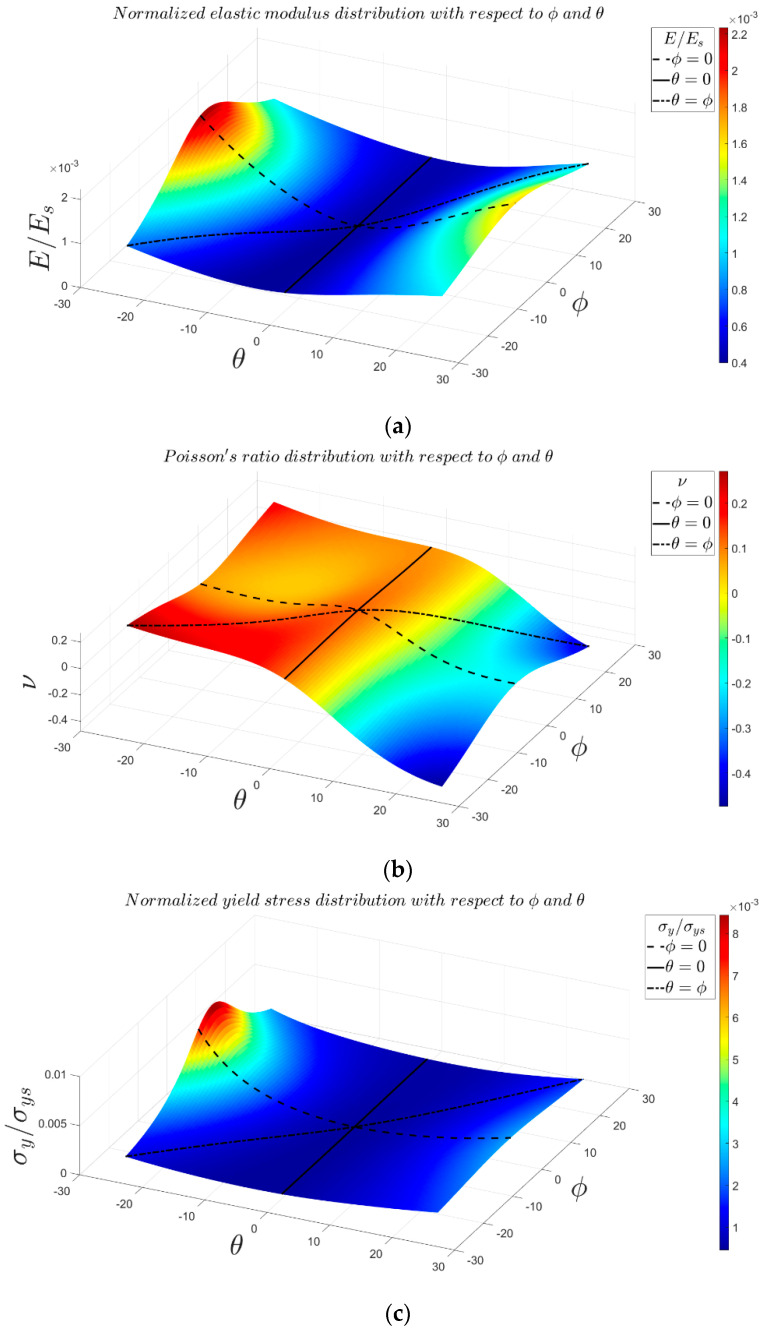
Variations of mechanical properties of general 3D re-entrant unit cell with respect to angles θ and φ for $r_{1}=r_{2}=r_{3}=0.14b$: (**a**) elastic modulus, (**b**) Poisson’s ratio, and (**c**) yield stress. Specific cases of θ=0°, φ=0°, and θ=φ  are marked with solid, dashed, and dashed-dot lines, respectively.

**Figure 13 materials-14-00993-f013:**
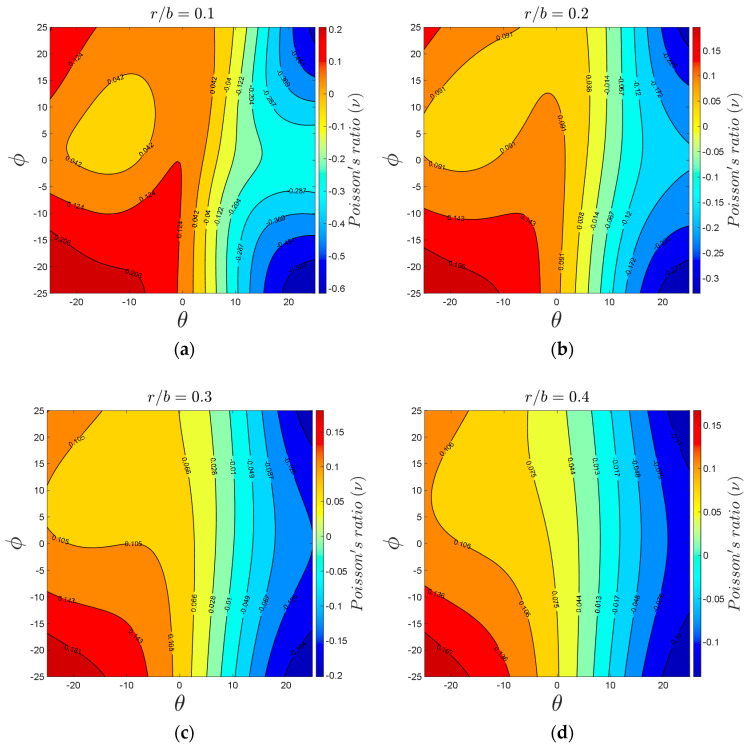
Variations of Poisson’s ratio with respect to θ and φ: (**a**) r/b=0.1, (**b**) r/b=0.2, (**c**) r/b=0.3, and (**d**) r/b=0.4.

**Figure 14 materials-14-00993-f014:**
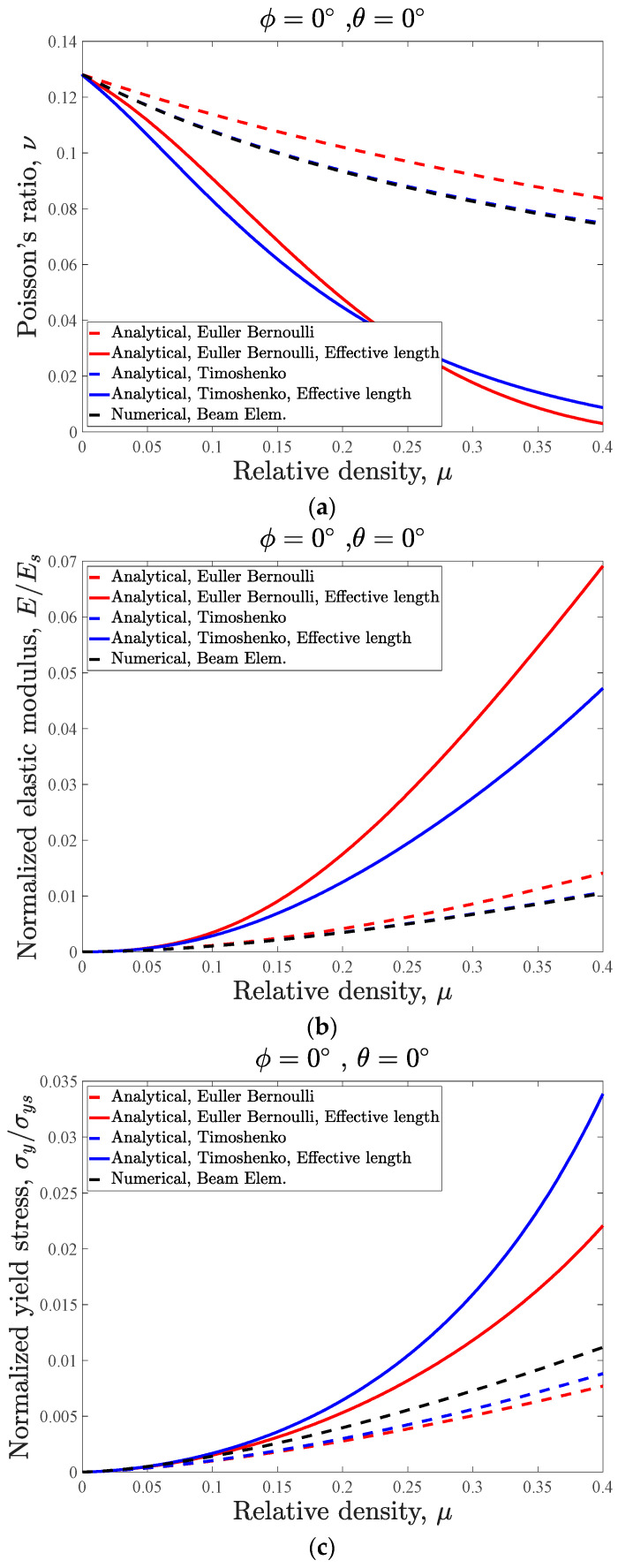
Mechanical properties variations of structure with φ=0, θ=0: (**a**) Poisson’s ratio, (**b**) elastic modulus, and (**c**) yield stress.

**Figure 15 materials-14-00993-f015:**
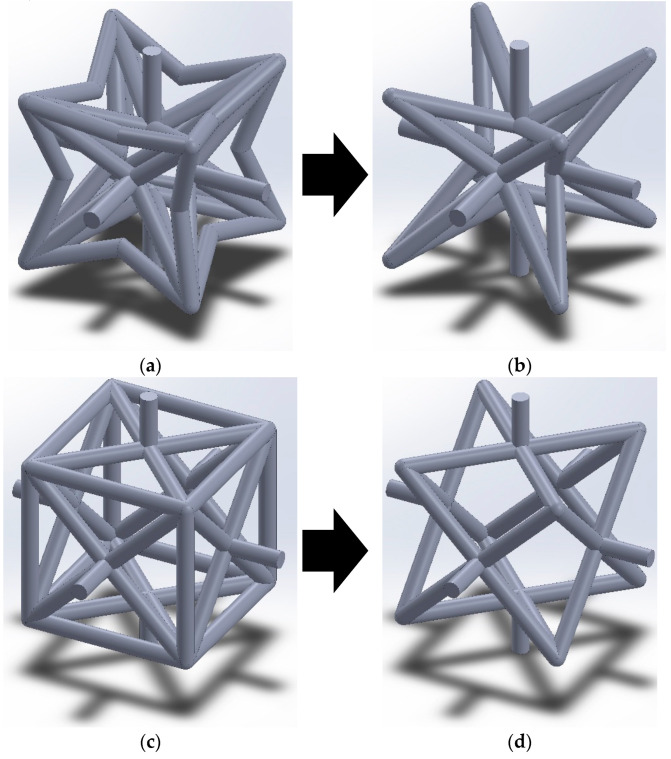
Generic geometry of the unit cell with positive (top), zero (middle), and negative (bottom) values of θ and φ for general unit cell (left column) and lightened unit cell (right column): (**a**) general re-entrant; (**b**) lightened re-entrant; (**c**) general zero Poisson’s ratio; (**d**) lightened zero Poisson’s ratio; (**e**) general convex; and (**f**) lightened convex unit cells.

**Figure 16 materials-14-00993-f016:**
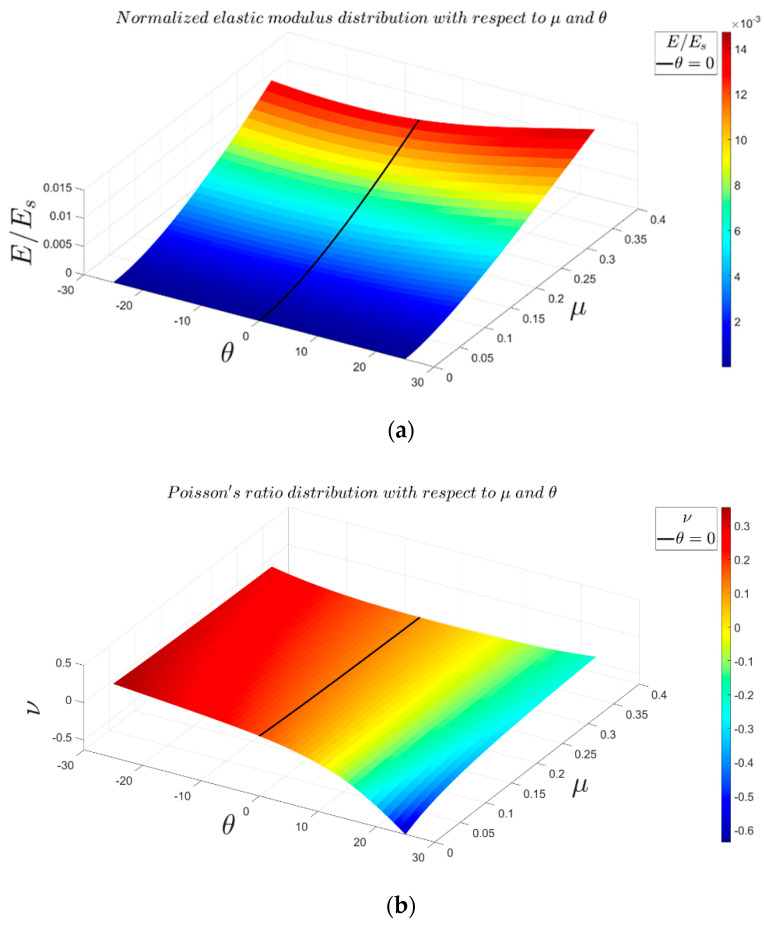
Variations of mechanical properties of lightened structure with respect to θ and μ: (**a**) elastic modulus, (**b**) Poisson’s ratio, and (**c**) yield stress. The specific case of θ=0 is shown as a solid line.

**Figure 17 materials-14-00993-f017:**
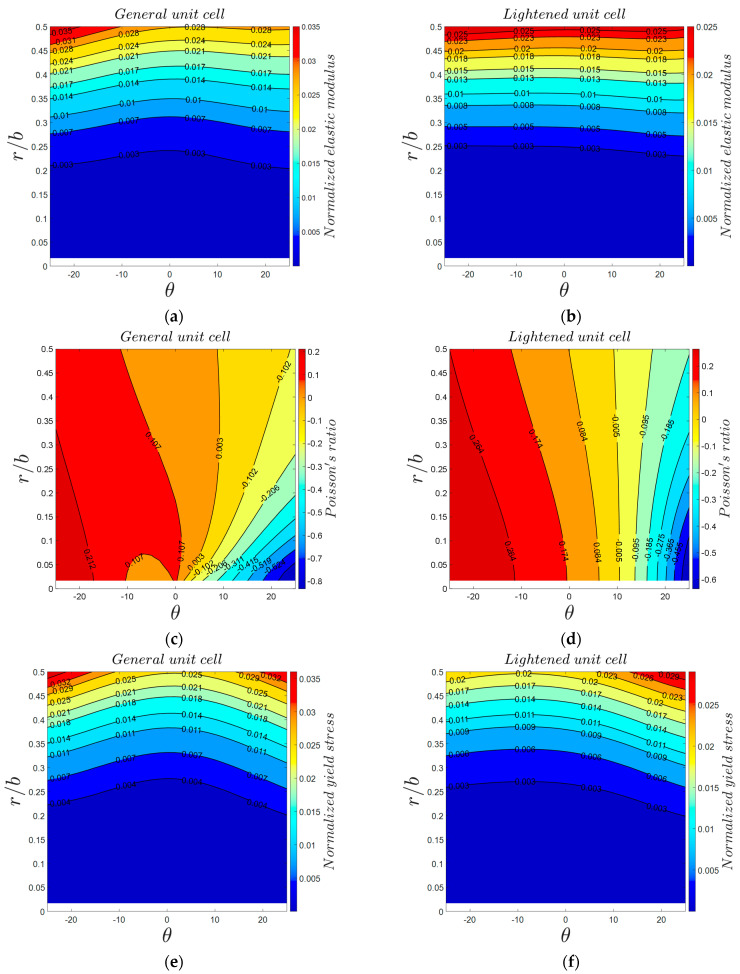
Contour plots of mechanical properties of general (left) and lightened (right) unit cells with respect to r/b and θ parameters: (**a**,**b**) normalized elastic modulus, (**c**,**d**) Poisson’s ratio, and (**e**,**f**) normalized yield stress.

**Figure 18 materials-14-00993-f018:**
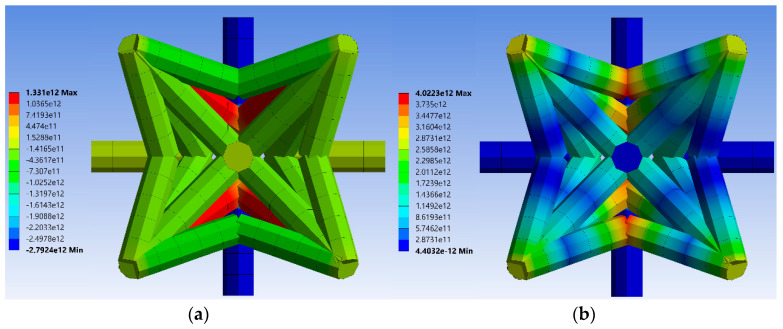
Stress distribution of FE model constructed by Beam 189 elements for μ=0.2: (**a**) normal stress and (**b**) flexural stress. Von-mises equivalent stress and strain, principal strains and stresses, normal and bending stress distributions are shown in Appendix A.

**Table 1 materials-14-00993-t001:** Parameters defined for summarizing the relationships for Euler–Bernoulli and Timoshenko beam theories.

Term	Euler-Bernoulli Theory	Timoshenko Theory
$S_{i}$	$\frac{A_{i}E_{s}}{l_{i}}$	$\frac{A_{i}E_{s}}{l_{i}}$
$W_{i}$	$\frac{G_{s}J_{i}}{l_{i}}$	$\frac{G_{s}J_{i}}{l_{i}}$
$T_{i}$	$\frac{12E_{s}I_{i}}{l_{i}^{3}}$	$\frac{1}{\frac{l_{i}^{3}}{12E_{s}I_{i}}+\frac{l_{i}}{\kappa A_{i}G_{s}}}$
$V_{i}$	$\frac{6E_{s}I_{i}}{l_{i}^{2}}$	$\frac{1}{\frac{l_{i}^{2}}{6E_{s}I_{i}}+\frac{2}{\kappa A_{i}G_{s}}}$
$U_{i}$	$\frac{4E_{s}I_{i}}{l}$	$\frac{\frac{2E_{s}I_{i}}{\kappa A_{i}G_{s}l}+\frac{2l_{i}}{3}}{\frac{l_{i}^{2}}{6E_{s}I_{i}}+\frac{2}{\kappa A_{i}G_{s}}}$

**Table 2 materials-14-00993-t002:** Elastic mechanical properties of cylindrical specimens made from poly-lactic acid (PLA) filaments.

Property	0°	45°	90°
Elastic modulus (GPa)	2.016	1.964	1.824
Yield stress (MPa)	74.67	73.6	63.67

## Data Availability

Detailed data is contained within the article or Appendix A. More data that support the findings of this study are available from the author Naeim Ghavidelnia upon reasonable request.

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
