# Peer review of "Idealized 3D Auxetic Mechanical Metamaterial: An Analytical, Numerical, and Experimental Study"

_materials, 2021, doi:10.3390/ma14040993_

Round 1

Reviewer 1 Report

Dear Authors,

It is well written paper, with very interesting research results. The topic is very imprtant in 3D printing and I strongly recommend this paper for publication in the Journal. I can only recommend to include in the introduction some paper about thin walled component manufactured by 3D pritning what can make this paper more valuable.

1.Tensile Strength Analysis of Thin-Walled Polymer Glass Fiber Reinforced Samples Manufactured by 3D Printing Technology, https://doi.org/10.3390/polym12122783

Kind regards,

Reviewer

Author Response

Dear reviewer,

Please see our response attached.

Kind regards,

Reza Hedayati

Reviewer 2 Report

This paper analytically, numerically, and experimentally studied an idealized 3D auxetic mechanical metamaterial. The author should address the following points:

  1. The structure studied is shown in Figure 2. The author did not state clearly if the structure is their original design or it is from some reference. If it is based on the reference, then the author should include the reason to study this metamaterial.
  2. In the analytical part, based on my understanding, the effective length is an important consideration. But the authors made more efforts on the 2.1.2 stiffness matrix and showed the complete derivation process. I did not see the relationship between the effective length and stiffness matrix. 
  3. In figure 9, the author compared analytical, numerical, and experimental results. It is not clear to me why the numerical result has a large difference from the experimental and analytical results. It seems the effective length is not considered in the simulation, but I could not find the reason in the paper. 
  4. In the conclusion, the author did not mention the application of this metamaterial.

Author Response

(The authors gave the same response as above.)

Reviewer 3 Report

Report on the manuscript “Idealized 3D auxetic mechanical metamaterial: an analytical, numerical, and experimental study” by N. Ghavidelnia, M. Bodaghi and R. Hedayati.

This work concerns analytical, numerical, and experimental study of 3D re-entrant structure. In this work, two 3D re-entrant structures are studied. The first one is a well-known 3D re-entrant structure that has been extensively studied before. For this structure, the author provided extensive analytical, numerical, and experimental studies which are valuable itself. In this work, the authors also introduced a second structure with ‘reduced’ unit cell (i.e. having less ribs) which seems to be not considered before. For latter structure the analytical and numerical studies were performed. The lack of experimental study for the reduced unit cell (called by the Authors as "lightened unit cell") does not disqualify this work from publication. The authors compare the elastic properties of both structures to each other. In my opinion, this is very interesting work but prior to acceptance, the authors should consider and respond to the remarks below.

1) The authors should clarify in the work when they deal with the metamaterial and when with the unit cell. The two terms are not interchangeable as it is in the present version of the submitted manuscript. Clearly, the experiment was performed only for a unit cell and not for the metamaterial.

2) From the above comment it follows the present title of the paper which may be misleading. It could be changed.

3) The authors made the comparison of elastic properties of two structures in 3D pictures in supplementary materials mostly. I think it is one of the most interesting results that, in my opinion, should be included to the main text. In general, 3D figures are nice but less informative than 2D plots. For this reason, it would be good if the authors could provide an additional figure in the main text for the comparison for both structures for fixed values of angels (? and ?) similar like it was for methods in Fig. 9 but structures.

4) It would be interesting if the authors could comment on the impact of the overlapping effect at the vertices on the Poisson's ratio in relation to the reduced structure. There are fewer connections there! How does this effect contribute to the overall change of the Poisson's ratio of the reduced structure?

5) Page 15 lines 356-357 one can read: “This phenomenon becomes of much more significance in auxetic materials, as they have very acute angles at the vertices (see for example Figure 2 and Figure 8)” This statement is not correct. It should be rather model or structure because use the term “materials” is too general here. For example, it completely does not apply to auxetic foams, polymers, or other models of molecular auxetic materials (see references below). In this context, it would be worth it for the readers if the authors could extend the introduction and discuss, or at least mention, other types of auxeticity (see references below). One of the first mechanical structure was proposed by Almgren [J. Elasticity 1985, see below], the first foam was presented by Lakes [Science 1987], and the first 2D molecular models (spontaneously forming auxetic phases) were studied by Wojciechowski [Mol. Phys 1987; Phys. Lett. A 1989]. The term auxetic in the scientific literature was introduced by Evans [Endeavour 1991]. More recent publications and some fundamental auxetic models/materials could be also mentioned. Below are some details regarding various papers on auxetics relevant to this work:
AN ISOTROPIC 3-DIMENSIONAL STRUCTURE WITH POISSON RATIO=-1; By: ALMGREN, RF; JOURNAL OF ELASTICITY Volume: 15 Issue: 4 Pages: 427-430 Published: 1985

CONSTANT THERMODYNAMIC TENSION MONTE-CARLO STUDIES OF ELASTIC PROPERTIES OF A TWO-DIMENSIONAL SYSTEM OF HARD CYCLIC HEXAMERS; By: WOJCIECHOWSKI, KW; MOLECULAR PHYSICS, Volume: 61, Issue: 5, Pages: 1247-1258, Published: AUG 10 1987

TWO-DIMENSIONAL ISOTROPIC SYSTEM WITH A NEGATIVE POISSON RATIO; By: WOJCIECHOWSKI, KW; PHYSICS LETTERS A Volume: 137 Issue: 1-2 Pages: 60-64 Published: MAY 1 1989;

AUXETIC POLYMERS - A NEW RANGE OF MATERIALS; By: EVANS, KE; ENDEAVOUR Volume: 15 Issue: 4 Pages: 170-174 Published: 1991;

A microscopic model of a negative Poisson's ratio in some crystals; By: Ishibashi, Y; Iwata, M; JOURNAL OF THE PHYSICAL SOCIETY OF JAPAN Volume: 69 Issue: 8 Pages: 2702-2703 Published: AUG 2000;

Searching for auxetics with DYNA3D and ParaDyn; By: Hoover, WG; Hoover, CG; PHYSICA STATUS SOLIDI B-BASIC SOLID STATE PHYSICS Volume: 242 Issue: 3 Pages: 585-594 Published: MAR 2005;

Inducing out-of-plane auxetic behavior in needle-punched nonwovens; By: Verma, Prateek; Shofner, Meisha L.; Lin, Angela; Wagner KB; Griffin AC; PHYSICA STATUS SOLIDI B-BASIC SOLID STATE PHYSICS Volume: 252 Issue: 7 Special Issue: SI Pages: 1455-1464 Published: JUL 2015;

Auxeticity enhancement due to size polydispersity in fcc crystals of hard-core repulsive Yukawa particles; By: Piglowski, PM; Narojczyk, JW; Wojciechowski, KW; Tretiakov, KV; SOFT MATTER Volume: 13 Issue: 43 Pages: 7916-7921 Published: NOV 21 2017

Negative-Poisson's-Ratio Materials: Auxetic Solids; By: Lakes, Roderic S.; ANNUAL REVIEW OF MATERIALS RESEARCH, VOL 47 Book Series: Annual Review of Materials Research Volume: 47 Pages: 63-81 Published: 2017;

Auxeticity in Metals and Periodic Metallic Porous Structures Induced by Elastic Instabilities By: Duc Tam Ho; Cao Thang Nguyen; Kwon, Soon-Yong; Kim SY; PHYSICA STATUS SOLIDI B-BASIC SOLID STATE PHYSICS Volume: 256 Issue: 1 Article Number: 1800122 Published: JAN 2019;

Observation of Squeeze-Twist Coupling in a Chiral 3D Isotropic Lattice, By: Li, JH; Ha, CS; Lakes, RS PHYSICA STATUS SOLIDI B-BASIC SOLID STATE PHYSICS Volume: 257 Issue: 10 Article Number: 1900140 Published: OCT 2020;

Auxetic, Partially Auxetic, and Nonauxetic Behaviour in 2D Crystals of Hard Cyclic Tetramers; By: Tretiakov, KV; Wojciechowski, KW; PHYSICA STATUS SOLIDI-RAPID RESEARCH LETTERS Volume: 14 Issue: 7 Article Number: 2000198 Published: JUL 2020

6) Page 24 lines 562-563, one can read: “A critical case can be considered for the re-entrant structure in which angles ? and ? are considered to be zero.” At ? = ? = 0, this is not the re-entrant structure anymore! The convex structure are presented in Fig. 15d and Fig. 15f. This should be amended.

Minor remarks
1) Page 5 line 108: I cannot find A3 in Fig. 2.
2) Abbreviations should be checked, e.g. in page 18 line 457 AM is used but it is not explained earlier. All abbreviations should be expanded at first use in the text.
3) Description of figures 9, 14 should be corrected. Instead “… (a) elastic modulus, (b) Poisson’s ratio, … ” should be “… (a) Poisson’s ratio, (b) elastic modulus, …”.
4) All references should be checked, e.g. in refs 22, 43 there are lacking volume and page numbers.

Author Response

(The authors gave the same response as above.)

Reviewer 4 Report

Dear authors, the study provides a useful analytical approach for auxetic periodic metamaterials. Nevertheless, some major issues must be resolved before publishing. The design of solid FE models must be better explained with additional figures to improve readability. Also, the discussion sections must reflect the study strengths and shortcomings together with reflection with current literature.

Introduction:

P2/L81: Provide references

P4/Figure 2: What does the double arrow mean for DOF q10? Please clarify.

P5/Eq(1): Perhaps using crossectional areas A1, A2 and A3 in Eq.1 would be more readable and smooth.

P5/L123: What is meant by matrix material? It is confusing as there is no mention above that it is a composite material. Perhaps something like a baseline or bulk material…

P6/L154: This must be better explained. The beam theories cannot be linear, but perhaps describe the force/displacement relation to be linear. Those two assumptions are also the limitations of the study and must be discussed later.

P14/L324: This is not clear. The detailed FE figures where the maximum stress should be shown. The implication for using VM stress as well, because the struct can buckle with compressive stress, which is not measurable by VM stress. It is unsure that to estimate yield stress is enough to pick vertices B only. It may require to analyse individual structs for different loading paths. Please be aware of such limitations and consider critical discussion.

P17/L415: How were the material constants chosen? It should be based on experimental baseline material for 3d printer.

P17/L415: This is confusing, applying this boundary conditions described is not equal to periodic boundary BC as the displacement u_below is not qual to u_top. Please explain why the non-periodic boundary conditions were chosen if the structure is periodic. Moreover, please add the figure with boundary conditions applied, scrolling up is highly unreadable for readers and reviewers.

P17/L425: Please provide the strategy of how the mesh size was chosen, the current description is insufficient. What is meant by adaptive mesh size? Was the mesh size estimated based on some posterior error estimator? Or was it only on geometry features and lengthscales? Modelling structs with solids requires a sufficient number of elements along the crosssection to prevent locking effects. Please provide the order of solid elements and mesh used for all solid models.

Discussion:

The discussions on study limitations and further improvements are missing. Please discuss:

linearity assumptions-implies only small deformation,

buckling effects in structures-implies sharp loose of elastic properties

printing orientation as it is known to influence mechanical properties of the printed specimen.

Limitation using common Cauchy continuum for structures involving rotational DOFs and relative density (https://doi.org/10.1007/s11012-019-01000-8)

Comparison your solid FE model with homogenisation approach

Author Response

(The authors gave the same response as above.)

Round 2

Reviewer 4 Report

Dear authors, the study was significantly improved and in the reviewer's opinion can be published.